# A Revenue Function for Comparison-Based Hierarchical Clustering

**Aishik Mandal**[*]                                                                      *jitaishik@iitkgp.ac.in*
*Centre of Excellence in Artificial Intelligence*
*Indian Institute of Technology Kharagpur*

**Michaël Perrot**[*]                                                                     *michael.perrot@inria.fr*
*Univ. Lille, Inria, CNRS, Centrale Lille, UMR 9189 - CRIStAL, F-59000 Lille, France*

**Debarghya Ghoshdastidar**[*]                                                           *ghoshdas@cit.tum.de*
*Technical University of Munich*
*School of Computation, Information and Technology*
*Munich Data Science Institute*

**Reviewed on OpenReview:** *https://openreview.net/forum?id=QzWr4w8PXx*

## Abstract

Comparison-based learning addresses the problem of learning when, instead of explicit features or pairwise similarities, one only has access to comparisons of the form: *Object A is more similar to B than to C*. Recently, it has been shown that, in Hierarchical Clustering, single and complete linkage can be directly implemented using only such comparisons while several algorithms have been proposed to emulate the behaviour of average linkage. Hence, finding hierarchies (or dendrograms) using only comparisons is a well understood problem. However, evaluating their meaningfulness when no ground-truth nor explicit similarities are available remains an open question.

In this paper, we bridge this gap by proposing a new revenue function that allows one to measure the goodness of dendrograms using only comparisons. We show that this function is closely related to Dasgupta's cost for hierarchical clustering that uses pairwise similarities. On the theoretical side, we use the proposed revenue function to resolve the open problem of whether one can approximately recover a latent hierarchy using few triplet comparisons. On the practical side, we present principled algorithms for comparison-based hierarchical clustering based on the maximisation of the revenue and we empirically compare them with existing methods.

## 1 Introduction

In the past decade, there has been an exponential growth in the scope of data science and machine learning in domains such as psycho-physics (Shepard, 1962; Stewart et al., 2005; Haghiri et al., 2020) or cultural psychology (Berenhaut et al., 2022), evolutionary biology (Foulds et al., 1979; Semple and Steel, 2003; Catanzaro, 2009), or crowd-sourcing (Heikinheimo and Ukkonen, 2013; Ukkonen, 2017) among others. A type of data that has recently gained some traction in these contexts is *comparisons* (Stewart et al., 2005; Agarwal et al., 2007), particularly in the form of:

**Triplet comparison**: Binary response to the query—is object $i$ more similar to object $j$ than to object $k$?

**Quadruplet comparison**: Binary response to the query —are objects $i$ and $j$ more similar to each other than objects $k$ and $l$?

---

[*]All authors contributed equally.

Comparisons have been used in the psycho-physics literature for more than 50 years since it is known that humans can provide relative measurements better than absolute ones (Shepard, 1962; Stewart et al., 2005). It led to a surge of popularity of comparisons in the context of crowdsourced data about objects that cannot be represented by Euclidean features, such as food (Wilber et al., 2014) or musical artists (Ellis et al., 2002), or objects for which humans cannot robustly estimate a pairwise similarity, for instance cars (Kleindessner and von Luxburg, 2017) or natural scenes (Heikinheimo and Ukkonen, 2013). The purpose of collecting comparisons is often to learn patterns in the objects, such as latent clusters, or use them for prediction, as in classification. Hence, there has been significant development of algorithms for comparison-based learning (Agarwal et al., 2007; Heikinheimo and Ukkonen, 2013; Haghiri et al., 2017; Kazemi et al., 2018; Perrot and von Luxburg, 2019).

The present paper focuses on comparison-based hierarchical clustering. Clustering refers to partitioning a dataset into groups of similar objects, while hierarchical clustering is the problem of finding partitions of the data at different levels of granularity. It is natural to wonder how one can group similar objects or find a hierarchy of groups when neither features nor pairwise similarities are available, and one only has access to triplet or quadruplet comparisons. For instance, the objects in the food dataset (Wilber et al., 2014) can be broadly categorised into 'sweets or desserts' and 'main or savoury dishes', but the latter can be further sub-divided into meat dishes, soups and others. Surprisingly, interest in comparison-based (hierarchical) clustering stemmed in 1970s when single linkage clustering gained popularity, and researchers realised that the method uses only ordinal informal instead of absolute values of pairwise similarities (Janowitz, 1971; Sibson, 1972; Janowitz, 1979). Around the same time, works also started on the *consensus tree problem*, that is, constructing trees (hierarchies) from given sub-trees or ordinal relations (Adams III, 1972; Aho et al., 1981). The problem has since evolved as an important topic in both computational biology and computer science, most notably addressing the question of phylogenetic tree reconstruction under triplet or other ordinal constraints Semple and Steel (2003); Wu (2004); Snir and Yuster (2011). More recently, Ghoshdastidar et al. (2019) re-discovered that single and complete linkage can be computed using only few *actively chosen* quadruplet comparisons. There are, however, limited practical settings where the learning algorithm can actively decide which comparisons should be queried, and the most relevant case is that of learning from a set of *passively collected* comparisons. This is certainly true in the known practical applications of comparison-based hierarchical clustering, such as (hierarchical) clustering of objects from crowd-sourced comparisons (Ukkonen, 2017; Kleindessner and von Luxburg, 2017), finding communities in languages or in cultural psychology (Berenhaut et al., 2022), and constructing relational database queries Aho et al. (1981) or phylogenetic trees (Semple and Steel, 2003).

One of the fundamental problems in hierarchical clustering is to evaluate the *goodness* of a hierarchy. This issue is obviously inherent to identifying *better* hierarchical clustering algorithms. In the phylogenetics literature, the optimal hierarchy problem typically corresponds to the *minimum evolution problem*, where, given a set of species and a pairwise distance matrix (representing evolutionary distance between species), the goal is to find a weighted tree with minimal total edge weights that preserve the evolutionary distances (Foulds et al., 1979; Catanzaro, 2009). A similar philosophy exists in the early works on hierarchical clustering, where an algorithm is judged to better if the ultrametric induced by the output tree is closer to the specified pairwise dissimilarities among the given objects (Janowitz, 1979). More recently, there has been efforts to mathematically quantify the goodness of a hierarchy in terms of certain *cost or revenue functions* (Dasgupta, 2016; Moseley and Wang, 2017; Wang and Wang, 2020). Such formulations have led to a plethora of new methods for hierarchical clustering that also come with worst-case approximation guarantees (Cohen-Addad et al., 2019; Charikar et al., 2019; Chatziafratis et al., 2021).

**Motivation for this work and our contributions.** The main motivation for this work stems from the lack of cost or revenue functions that can be used in the comparison-based framework. Available goodness measures for trees can only be defined using pairwise (dis)similarities (Dasgupta, 2016; Moseley and Wang, 2017; Wang and Wang, 2020). Hence, existing works on comparison-based hierarchical clustering either demonstrate the meaningfulness of the computed hierarchies visually or in artificial settings, where comparisons are derived from pairwise similarities (Kleindessner and von Luxburg, 2017; Ghoshdastidar et al., 2019). Neither solution is useful in practice, where one only has access to comparisons. In this paper, we propose new revenue functions for dendrograms that are only based on triplet or quadruplet comparisons (Section

4). We show that the proposed comparison-based revenues are equivalent to Dasgupta's cost or revenue (Dasgupta, 2016; Moseley and Wang, 2017) applied to particular pairwise similarities that can be computed from comparisons. Interestingly, the pairwise similarities that arise from this equivalence are known in the comparison-based clustering literature (Perrot et al., 2020).

Section 5 demonstrates that the proposed revenue function meaningfully captures the goodness of a hierarchical tree. For this purpose, we consider the problem of reconstructing a latent hierarchy (for example, a phylogeny tree) from ordinal constraints (Emamjomeh-Zadeh and Kempe, 2018). In particular, we show that, when all possible triplets among the objects are available, the dendrogram corresponding to the latent hierarchy maximises the proposed revenue function. This, in turn, implies that one can mathematically formulate the triplet-based hierarchical clustering problem as a *maximum triplet comparison revenue problem*. We further address the question of whether one can approximately recover the latent hierarchy using fewer than $\Omega(n^3)$ triplets. This problem has not been directly addressed in previous works (see Section 2). We show that only $O(n^2 \log n / \epsilon^2)$ passive triplets suffice to obtain a $(1 - \epsilon)$-approximation of the optimal revenue.

Finally, Sections 6–7 use the connection of the proposed revenue functions to the *additive similarities* in Perrot et al. (2020) to present two variants of average linkage hierarchical clustering based on passive triplet or quadruplet comparisons. The performance of these approaches is empirically compared with state of the art baselines using synthetic and real datasets.

## 2 Related Work

In this section, we briefly review the algorithmic developments of comparison-based hierarchical clustering, as well as existing theoretical results related to this problem. As noted earlier, interest in comparison-based hierarchical clustering stemmed from different applications. The current literature consists of two lines of research—works related to reconstruction of phylogenetic trees (Wu, 2004; Snir and Yuster, 2011; Chatziafratis et al., 2021) and those focusing on ordinal data analysis from crowd-sourced data (Kleindessner and von Luxburg, 2017; Ghoshdastidar et al., 2019).

In ordinal data analysis literature, the most widely used principle is that of ordinal embedding, where the underlying idea is to retrieve Euclidean representations of the objects that respect the available comparisons as well as possible (see the review in Vankadara et al. (2019) for more details). The embedded data can be subsequently used for (hierarchical) clustering. While this principle provides flexibility in the choice of clustering methods, the Euclidean restriction of the underlying data often leads to inaccurate representations, and hence, poor performance in the context of hierarchical clustering (Ghoshdastidar et al., 2019). The restrictive assumption of Euclidean embedding is avoided by computing pairwise similarities from available comparisons (Kleindessner and von Luxburg, 2017; Ghoshdastidar et al., 2019). Standard hierarchical clustering algorithms, such as average linkage, can then be applied using the pairwise similarities.

An alternative approach for comparison-based (hierarchical) clustering is to define an appropriate cost or objective based on comparison and directly optimise it. Ukkonen (2017) employs such a technique for clustering using crowd-sourced data, while this principle underlies most techniques in consensus tree problems or phylogenetic tree reconstruction. In the latter context, two well-studied optimisation problems are *maximum rooted triplet consistency* (Wu, 2004; Byrka et al., 2010)—finding a hierarchy that satisfies most, if not all, given triplets—and *maximum quartet consistency* (Snir and Yuster, 2011; Jiang et al., 2000)—where one has access to quartets (sub-trees with four leaves indicating which pairs should be merged first) and the problem is to find a tree that satisfies most given quartets.[*] Other related optimisation problems as well as various constraints other than triplets or quartets have been studied (Snir and Rao, 2010; Chatziafratis et al., 2021). Since the focus of the present paper is to define a revenue for trees (see Section 4), our work naturally belongs to this broad class of hierarchical clustering algorithms based on revenue maximisation. However, in Theorem 1, we relate the proposed revenues to pairwise similarities computed from comparisons. Hence,

---

[*]Note that quartets are different from quadruplets though both are defined on four objects. More precisely, a quartet on $i, j, k, l$ corresponds to information that $i, j$ and $k, l$ should be merged in the tree before all four are merged. Using the notation from Section 3, a quadruplet $(i, j, k, l)$ only implies $s_{ij} > s_{kl}$ whereas a quartet on $i, j, k, l$ implies $\min\{s_{ij}, s_{kl}\} > \max\{s_{ik}, s_{il}, s_{jk}, s_{jl}\}$.

the present paper connects the optimisation principle to the aforementioned approach of defining pairwise similarities from comparisons.

Prior works on comparison-based hierarchical clustering provide a range of computational and statistical results. On the computational side, it is known that both the problems of maximum rooted triplet consistency and maximum quartet consistency are NP-hard (Byrka et al., 2010; Snir and Yuster, 2011). However, polynomial-time constant factor approximation algorithms are known in both cases, assuming that a uniformly random subset of triplets/quartets is available. For triplets, Wu (2004) provides a $\frac{1}{3}$-approximation algorithm—a fraction at least $\frac{1}{3}$ of the given triplets are satisfied—which is slightly improved in Byrka et al. (2010). For quartets, polynomial-time algorithms that satisfy at least a $(1 - \epsilon)$-fraction of the given quartets are known (Jiang et al., 2000; Snir and Yuster, 2011). While the above results focus on finding hierarchies that only match the given triplets/quartets, Emamjomeh-Zadeh and Kempe (2018) show that the true (latent) hierarchy can be recovered only if $\Omega(n^3)$ passive (uniformly sampled) triplets are available. In contrast, only $O(n \log n)$ triplets suffice if they are actively queried. In Section 5, we show that only $O(n^2 \log n / \epsilon^2)$ uniformly sampled triplets suffice to obtain a $(1 - \epsilon)$-approximation of the optimal triplet revenue.

A different latent model is considered in Ghoshdastidar et al. (2019) and Perrot et al. (2020), where the objects have latent (noisy) pairwise similarities that have a (hierarchical) cluster structure. Noisy triplets/quadruplets are uniformly sampled following the noisy latent similarities. While Ghoshdastidar et al. (2019) focus on quadruplet-based hierarchical clustering and show that $O(n^{3.5} \log n)$ suffice to recover the latent hierarchy, Perrot et al. (2020) show that flat latent clusters can be exactly recovered using only $O(n^2 \log n)$ uniformly sampled triplets/quadruplets. Although our model is different from Perrot et al. (2020), we obtain a similar $O(n^2 \log n)$ upper bound on sample complexity—even when the triplets are noisy.

## 3  Preliminaries

We consider the problem of hierarchical clustering of a set of $n$ objects, denoted by $[n] = \{1, 2, \ldots, n\}$. In this paper, we assume that a hierarchy or dendrogram on $[n]$ is a binary tree $H$ whose root node is the set $[n]$, each leaf node is a singleton containing one of the $n$ objects, and each internal node represents a set $C \subseteq [n]$ with its two children, $C_1$ and $C_2$, denoting a partition of $C$, that is $\min(|C_1|, |C_2|) > 0$, $C = C_1 \cup C_2$, and $C_1 \cap C_2 = \emptyset$. In the following, we use binary tree or tree to designate a hierarchy. For node $C$, we use $H(C)$ to denote the subtree rooted at $C$ and $|H(C)|$ represents the number of leaves in the subtree, or equivalently, the number of objects in the set $C$. For objects $i, j \in [n]$, let $i \vee j$ denote the smallest node in the tree containing both $i$ and $j$, and $H(i \vee j)$ denote the smallest subtree containing both $i$ and $j$. The goal of hierarchical clustering is to find a dendrogram $H$ that is *optimal*, or at least *good*, in some sense. In the next subsections, we recall Dasgupta's cost for hierarchical clustering that allows one to measure the goodness of a dendrogram given full access to pairwise similarities, and then describe the comparison-based learning framework, where only triplet or quadruplet comparisons are available. In the paper, we use the standard Landau notations $O(\cdot)$, $\Omega(\cdot)$, $o(\cdot)$, where the asymptotics are defined with respect to $n$.

### 3.1  Dasgupta's Cost for Hierarchical Clustering

Suppose one has access to a function $s : [n] \times [n] \to \mathbb{R}$, symmetric, such that $s_{ij} = s(i, j)$ denotes the pairwise similarity between objects $i, j \in [n]$. Dasgupta's cost function (Dasgupta, 2016) for a dendrogram $H$ on $[n]$, with respect to the pairwise similarity $s$, is defined as

$$Dcost(H, s) = \sum_{i, j \in [n], \ i < j} s_{ij} \cdot |H(i \vee j)|. \tag{1}$$

An equivalent definition of the above cost can be found in Wang and Wang (2020), where the cost is expressed in terms of triplets of objects instead of pairs. To find a good dendrogram, Dasgupta (2016) proposed to minimize this cost over all trees. While it is NP-hard to find the optimal solution, several relaxations are known to have constant factor approximation guarantees for Dasgupta's cost or related quantities. In

particular, Moseley and Wang (2017) defined Dasgupta's revenue function,

$$Drev(H,s) = n \sum_{i,j \in [n], \ i<j} s_{ij} - Dcost(H,s). \tag{2}$$

Note that since $\sum s_{ij}$ is fixed, maximising $Drev(H,s)$ over all binary trees is equivalent to minimising $Dcost(H,s)$. It can then be shown that the revenue of the tree $H$ obtained from average linkage achieves a revenue $Drev(H,s)$ that is at least $\frac{1}{3}$ of the revenue of the optimal tree that would achieve the best revenue, provided that the similarity function $s$ is non-negative.

### 3.2 Comparison-based Learning

In the present paper, we assume that the pairwise similarities $\{s_{ij}\}_{i,j \in [n]}$ are not available. Instead the algorithm has access to either a set of triplets $\mathcal{T}$, which is a subset of

$$\mathcal{T}_{all} = \{(i,j,k) \in [n]^3 : s_{ij} > s_{ik}, \ i,j,k \text{ distinct}\},$$

or a set of quadruplets $\mathcal{Q} \subseteq \mathcal{Q}_{all}$, where

$$\mathcal{Q}_{all} = \{(i,j,k,l) \in [n]^4 : \ s_{ij} > s_{kl}, i < j, \ k < l, \ (i,j) \neq (k,l)\}.$$

Since the pairwise similarities are assumed symmetric, we set $i < j$ and $k < l$ to avoid considering the same comparison multiple times. We note that the number of possible comparisons is high—$|\mathcal{Q}_{all}| = O(n^4)$ and $|\mathcal{T}_{all}| = O(n^3)$—but, in practice, the observed comparisons, $\mathcal{T}$ or $\mathcal{Q}$ may be fewer than that, about $O(n^2)$ comparisons, as can be seen from Table 3. Note that whenever a triple $(i,j,k)$ is considered, $\mathcal{T}$ contains either $(i,j,k)$ or $(i,k,j)$, depending on whether $s_{ij} \lessgtr s_{ik}$. The same holds for quadruplets. We further assume that the observed set of comparisons $\mathcal{T}$, or $\mathcal{Q}$, is passively collected, that is the algorithm cannot decide which comparisons should be present in it—as opposed to the active setting where the algorithm can choose which comparisons should be observed (Ghoshdastidar et al., 2019).

## 4 Comparison-based Revenue

We present two comparison-based revenue functions for hierarchical clustering, one in the triplets framework and the other for quadruplet comparisons.

**Triplet comparisons.** We first consider the case of triplets and assume that the algorithm has access to a passively collected set of triplets $\mathcal{T}$. We define the triplet comparison revenue of a binary tree (dendrogram) $H$ on $[n]$, using triplets $\mathcal{T}$, as

$$Trev(H, \mathcal{T}) = \sum_{(i,j,k) \in \mathcal{T}} \Big( |H(i \vee k)| - |H(i \vee j)| \Big). \tag{3}$$

For every $(i,j,k) \in \mathcal{T}$, we know that $i$ is more similar to $j$ than to $k$, and hence, we prefer to merge $i,j$ before merging $i$ and $k$. It means the ideal tree should have $|H(i \vee k)| > |H(i \vee j)|$ for every $(i,j,k) \in \mathcal{T}$. Hence, it is desirable to maximise $|H(i \vee k)| - |H(i \vee j)|$ for every observed triplet $(i,j,k) \in \mathcal{T}$. We then propose to formulate triplets comparison-based hierarchical clustering as the problem of maximizing $Trev(H, \mathcal{T})$ over all binary trees.

**Remark 1.** *We note that the proposed triplet revenue is significantly different from the triplet based cost presented in Wang and Wang (2020). The most important distinction is that the triplet cost in Wang and Wang (2020) is a reformulation of Dasgupta's cost, and requires knowledge of pairwise similarities. In contrast, the revenue in equation 3 is computed only from triplet comparisons without access to pairwise similarities.*

**Quadruplet comparisons.** The above formulation can be similarly stated in the quadruplets setting. Assuming that the algorithm has access to a passively collected set of quadruplets $\mathcal{Q}$, we define the quadruplet comparison revenue of a binary tree $H$ on $[n]$ as

$$Qrev(H, \mathcal{Q}) = \sum_{(i,j,k,l) \in \mathcal{Q}} \Big( |H(k \vee l)| - |H(i \vee j)| \Big). \tag{4}$$

Similar to the triplet setting, every $(i, j, k, l) \in \mathcal{Q}$ indicates that $i, j$ should be merged earlier than $k, l$ in $H$, and we prefer trees such that $|H(k \vee l)| \geq |H(i \vee j)|$. We propose to achieve this by finding a tree that maximises $Qrev(H, \mathcal{Q})$.

**Connection with Dasgupta's cost.** While one may try to directly maximise the above comparison-based revenue functions, the following equivalence to Dasgupta's cost and revenue allows us to employ existing methods for hierarchical clustering that require pairwise similarities. In the following, let $\mathbb{I}_E$ denote the indicator of event $E$, that is, $\mathbb{I}_E = 1$ if $E$ happens, and 0 otherwise.

**Theorem 1.** *For any given set of triplets $\mathcal{T}$ and any dendrogram $H$ on $[n]$,*

$$Trev(H, \mathcal{T}) = -Dcost(H, s^{AddS3}) = Drev(H, s^{AddS3}),$$

*where $s^{AddS3}$ refers to the additive similarity from triplets (AddS3) defined by Perrot et al. (2020)*

$$s_{ij}^{AddS3} = \sum_{k \neq i,j} \Big( \mathbb{I}_{(i,j,k) \in \mathcal{T}} - \mathbb{I}_{(i,k,j) \in \mathcal{T}} + \mathbb{I}_{(j,i,k) \in \mathcal{T}} - \mathbb{I}_{(j,k,i) \in \mathcal{T}} \Big)$$

*Similarly, for any set of quadruplets $\mathcal{Q}$ and dendrogram $H$,*

$$Qrev(H, \mathcal{Q}) = -Dcost(H, s^{AddS4}) = Drev(H, s^{AddS4}),$$

*where $s^{AddS4}$ is the additive similarity from quadruplets (AddS4) defined by Perrot et al. (2020)*

$$s_{ij}^{AddS4} = \sum_{k \neq l, \ (k,l) \neq (i,j)} \Big( \mathbb{I}_{(i,j,k,l) \in \mathcal{Q}} - \mathbb{I}_{(k,l,i,j) \in \mathcal{Q}} \Big).$$

*Proof idea (details in appendix).* Proving $Trev(H, \mathcal{T}) = -Dcost(H, s^{AddS3})$ involves a rearrangement of terms, with the observation that, for every $i, j$, the term $|H(i \vee j)|$ appears in the summation in equation 3 with coefficient $-1$ when $(i, j, k) \in \mathcal{T}$ or $(j, i, k) \in \mathcal{T}$ and with coefficient $+1$ when $(i, k, j) \in \mathcal{T}$ or $(j, k, i) \in \mathcal{T}$. Adding these coefficients for all $k \neq i, j$ gives us $-s_{ij}^{AddS3}$, and proves the equality. The second equality $-Dcost(H, s^{AddS3}) = Drev(H, s^{AddS3})$ simply follows from the observation that $\sum_{i<j} s_{ij}^{AddS3} = 0$. The proof for quadruplets is similar. $\qquad\square$

## 5 Recovering a Latent Hierarchy by Triplet Revenue Maximisation

In this section, we consider the problem of recovering a latent hierarchy from triplet comparisons, earlier studied in Emamjomeh-Zadeh and Kempe (2018). Let $H_0$ be a hierarchy on $[n]$, from which we derive a set of triplets[*]

$$\mathcal{T}_0 = \{(i, j, k), (j, i, k) \ : \ |H_0(i \vee j)| < \min(|H_0(i \vee k)|, |H_0(j \vee k)|)\}. \tag{5}$$

One can show that any rooted tree $H'$ that satisfies all triplets in $\mathcal{T}_0$ is equivalent to $H_0$, up to isomorphic transformations, and hence, one can exactly recover $H_0$ given $\mathcal{T}_0$. Note that $|\mathcal{T}_0| = n\binom{n}{2}$. It is natural to ask whether, $H_0$ can be recovered if a significantly smaller number of triplets are observed. To this end, Emamjomeh-Zadeh and Kempe (2018) show that if the algorithm can choose the triplets to be queried (active setting), then a deterministic algorithm can recover $H_0$ using only $n \log_2 n$ queries. However, the authors also construct a (randomised) $H_0$ to show that for any fixed set $\mathcal{T} \subset \mathcal{T}_0$ with fewer than $n^3/48$ triplets, the

---

[*]Emamjomeh-Zadeh and Kempe (2018) consider triples of the form $\{i, j, k\}$ that imply $i, j$ are closer to each other than $k$, with respect to $H_0$. Each such triple $\{i, j, k\}$ correspond to two triplets $(i, j, k)$ and $(j, i, k)$ in our setting.

latent hierarchy $H_0$ cannot be recovered with probability at least $1/2$. This raises the question—can one approximately recover $H_0$ from a smaller set of triplets $\mathcal{T}$?

We use the proposed triplets-based revenue to answer this question in the affirmative. Before providing an approximation guarantee, we first show the significance of our formulation in this context by proving that one can recover $H_0$ from $\mathcal{T}_0$ by maximising $Trev$.

**Proposition 2.** *Consider the aforementioned setting, where $H_0$ is a hierarchy on $[n]$ objects, and $\mathcal{T}_0$ is the corresponding set of triplets as defined above. Then*

$$H_0 = \arg\max_{H} \ Trev(H, \mathcal{T}_0),$$

*where the maximisation is over all binary trees $H$ on $[n]$.*

*Proof idea (details in appendix).* One can show that if $\mathcal{T}_0$ is a set of triplets corresponding to hierarchy $H_0$ according to equation 5 and $(s_{ij})_{1 \leq i,j \leq n}$ denote the AddS3 similarity derived from $\mathcal{T}_0$ (cf. Theorem 1), then $s_{ij} = 2n + 2 - 3|H_0(i \vee j)|$, that is, one can recursively construct $H_0$ from the AddS3 similarities. We further use this representation to show that if $\mathcal{T}_0, \mathcal{T}_1$ are respectively derived from two hierarchies $H_0, H_1$ according to equation 5, then there is a symmetry in the triplet revenue of the form $Trev(H_0, \mathcal{T}_1) = Trev(H_1, \mathcal{T}_0)$.

The above symmetry implies that proving $H_0$ uniquely maximises $Trev(H, \mathcal{T}_0)$ is equivalent to showing that $Trev(H_0, \mathcal{T}_0) > Trev(H_0, \mathcal{T}_1)$ for every triplet set $\mathcal{T}_1$ that correspond to a binary tree $H_1$ on $[n]$ that is not isomorphic to $H_0$. This last claim can be proved by showing that every $i, j, k$—for which the ordering of merger is changed between $H_0$ and $H_1$—has a positive contribution in $Trev(H_0, \mathcal{T}_0) - Trev(H_0, \mathcal{T}_1)$. $\qquad \square$

Emamjomeh-Zadeh and Kempe (2018) prove the uniqueness of the hierarchy that satisfies all the triplets in $\mathcal{T}_0$, that is, $H_0$ maximises the function $f(H, \mathcal{T}_0) = \sum_{(i,j,k) \in \mathcal{T}_0} \mathbb{I}_{|H(i \vee k)| > |H(i \vee j)|}$. While maximising $Trev(H, \mathcal{T}_0)$ seems to be a relaxation of maximising $f(H, \mathcal{T}_0)$ in this context, Proposition 2 shows that both problems have the same optimal solution $H_0$.

### 5.1 Approximate Recovery of $H_0$ Using Passive Triplets

We consider the setting, where $\mathcal{T}_0$ is not completely available but one has access to a uniformly sampled subset $\mathcal{T} \subseteq \mathcal{T}_0$. We show that $|\mathcal{T}| = O(n^2 \log n / \epsilon^2)$ triplets suffice to obtain a tree $\widehat{H}$ such that $Trev(\widehat{H}, \mathcal{T}_0) \geq (1 - \epsilon) \cdot Trev(H_0, \mathcal{T}_0)$, that is, we get a good approximation of $H_0$ with much fewer than $n^3$ samples, although we may not exactly recover $H_0$. We consider the following uniform sampling to obtain $\mathcal{T}$. Let $p_n \in (0, 1]$ denote a sampling probability, depending on $n$. For every pair of triplets $(i, j, k), (j, i, k) \in \mathcal{T}_0$, we add the pair to $\mathcal{T}$ with probability $p_n$. We state the following approximation guarantee for trees derived using $\mathcal{T}$.

**Theorem 3.** *For a triplet set $\mathcal{T}$ obtained from the above sampling procedure, consider the hierarchy*

$$\widehat{H} = \arg\max_{H} \ Trev(H, \mathcal{T}).$$

*For any constants $\alpha > 0$ and $0 < \epsilon < 1/2$, if $n > 8/\epsilon$ and $p_n > 2^{12} \cdot (\alpha + 2) \log n / n\epsilon^2$, then with probability at least $1 - 2n^{-\alpha}$,*

$$0.1 p_n n^3 \leq |\mathcal{T}| \leq 0.5 p_n n^3 \qquad and \qquad Trev(\widehat{H}, \mathcal{T}_0) \geq (1 - \epsilon) \cdot Trev(H_0, \mathcal{T}_0). \tag{6}$$

*Proof idea (details in appendix).* We need to relate $Trev(H, \mathcal{T})$ with $Trev(H, \mathcal{T}_0)$ for every tree $H$. Since, the revenue function is linear with respect to the observed triplets, $\mathbb{E}[Trev(H, \mathcal{T})] = p_n \cdot Trev(H, \mathcal{T}_0)$, where the expectation is with respect random observation of each triplet pair in $\mathcal{T}_0$. We use concentration inequalities to bound deviation from expectation, $\max_H |Trev(H, \mathcal{T}) - \mathbb{E}[Trev(H, \mathcal{T})]|$. Although there are exponentially many $H$, one can note that, due to Theorem 1, the deviation can be controlled by bounding the maximum deviation of the $\binom{n}{2}$ AddS3 similarities, $\max_{i<j} |s_{ij} - \mathbb{E}[s_{ij}]|$. For this we use, Bernstein inequality (for each $s_{ij}$) followed by union bound (over all $i, j$). Thus, we show that for any constant $\alpha > 0$, if $p_n > (\alpha + 2) \log n / n$, then $\max_H |Trev(H, \mathcal{T}) - p_n Trev(H, \mathcal{T}_0)| = O\left(n^3 \sqrt{p_n n \log n}\right)$ with probability

$1 - n^{-\alpha}$. Using concentration for both $H_0, \widehat{H}$, and noting that $Trev(\widehat{H}, \mathcal{T}) \geq Trev(H_0, \mathcal{T})$, we have that $Trev(\widehat{H}, \mathcal{T}_0) \geq Trev(H_0, \mathcal{T}_0) - O\left(\sqrt{\frac{n^7 \log n}{p_n}}\right)$. For stated condition on $p_n$, the second term is at least $\frac{\epsilon n^4}{24}$. Next, we show that $Trev(H_0, \mathcal{T}_0) \geq \frac{n^4}{12} - \frac{2(n^3 - n^2 - n)}{3}$, which is at least $(1 - \epsilon)\frac{n^4}{12}$ for $n > 8/\epsilon$. This follows since AddS3 similarities computed from $\mathcal{T}_0$ are of the form $s_{ij} = 2n + 2 - 3|H_0(i \vee j)|$, and hence, we can rewrite $Trev(H_0, \mathcal{T}_0)$ in terms sizes of internal nodes. The lower bound follows from inductive arguments. Combining the lower bound with the deviation bound results in the theorem. The claim $|\mathcal{T}| = \Theta(p_n n^3)$ with probability $1 - n^{-\alpha}$ follows from $\mathbb{E}|\mathcal{T}| = p_n|\mathcal{T}_0| = \Theta(p_n n^3)$ and multiplicative Chernoff inequality. □

With $p_n$ fixed at the stated threshold, Theorem 3 shows that, with probability $1 - 2n^{-\alpha}$, we can achieve $(1 - \epsilon)$-approximation of triplet revenue using $|\mathcal{T}| = \Theta(n^2 \log n/\epsilon^2)$ triplets. The result in Emamjomeh-Zadeh and Kempe (2018, Proposition 2.2)—that $\Omega(n^3)$ triplets are necessary to exactly recover $H_0$—hinges on the fact that it is impossible to correctly guess the hierarchy at the lowest level of the tree $H_0$ using fewer comparisons. Since errors in the lowest level do not significantly affect $Trev$, we can achieve the $(1 - \epsilon)$-approximation in Theorem 3. However, note that $\widehat{H}$ may not be efficiently computable as it requires exhaustive search over all trees. We discuss practical algorithms in the next section.

Theorem 3 is stated in the noiseless setting, where it is assumed that every observed triplet in $\mathcal{T}$ is correct. It is natural to ask if Theorem 3 still holds under a noisy setting, where some triplets may be flipped with some probability. To formalise this, let $\mathcal{T} \subseteq \mathcal{T}_0$ be a set of triplets obtained from the sampling procedure in Theorem 3. Let $\mathcal{T}'$ be constructed such that, for every $(i, j, k) \in \mathcal{T}$, $\mathcal{T}'$ contains $(i, j, k)$ with probability $1 - \delta$, or $(i, k, j)$ with probability $\delta$. The random flipping of labels is independent for all $(i, j, k) \in \mathcal{T}$. We obtain the following corollary from a minor modification of the above proof (details in appendix).

**Corollary 4.** *For any fixed flipping probability $\delta \in (0, \frac{1}{2})$, and for any $\alpha > 0$ and $0 < \epsilon < 1/2$, $\max_H Trev(H, \mathcal{T}') \geq (1 - \epsilon) \cdot Trev(H_0, \mathcal{T}_0)$ with probability $1 - n^{-\alpha}$ if $n > 8/\epsilon$ and $p_n > \frac{2^{12} \cdot (\alpha+2) \log n}{n\epsilon^2(1-2\delta)^2}$.*

## 6 Comparison-based Algorithms for Hierarchical Clustering

The equivalence between comparison-based revenues and Dasgupta's revenue, stated in Theorem 1, implies that one may simply employ standard hierarchical clustering algorithms using the pairwise similarities AddS3 or AddS4, depending on whether one has access to triplets or quadruplets. This makes it possible to use the well-established literature on hierarchical clustering with pairwise similarities. In fact, as mentioned before, previous works on passive comparison-based hierarchical clustering also follow this philosophy using other kind of pairwise similarities obtained from the comparisons (Kleindessner and von Luxburg, 2017; Ghoshdastidar et al., 2019). Unlike previous works, our use of AddS3 or AddS4 stems from a revenue maximisation formulation that allows us to consider an approach based on the average linkage (AL) clustering algorithm, that is, the following procedure:

| | |
|---|---|
| **AddS3-AL (or AddS4-AL)** | |
| **Given.** | A set of triplets $\mathcal{T}$ (or quadruplets $\mathcal{Q}$) on $[n]$ |
| **Step 1.** | Compute the pairwise similarity function $s^{AddS3}$ (or $s^{AddS4}$) for every pair of objects |
| **Step 2.** | Run average linkage algorithm with $s^{AddS3}$ (or $s^{AddS4}$) |
| **Output.** | The tree or dendrogram $H$ on the $n$ objects |

**Remark on approximation guarantee.** Average linkage enjoys theoretical guarantees under the assumption that the similarities are always positive. Moseley and Wang (2017) show that average linkage achieves a worst-case $\frac{1}{3}$-approximation for revenue maximisation. Unfortunately, this result does not readily extend to AddS3-AL and AddS4-AL, as these similarities may be negative in some cases. A possible approach could be to add a positive constant to all the similarities to ensure that they are positive. Although this does not change the optimal tree or the one obtained from average linkage, a $\frac{1}{3}$-approximation for the modified revenues (considering revised similarities) does not imply a $\frac{1}{3}$-approximation for the original revenues.

Based on the proof of Moseley and Wang (2017), one can show that AddS3-AL (or AddS4-AL) returns a tree with non-negative triplet (or quadruplet) comparison revenue. Whether approximation guarantees may also be derived for AddS3-AL and AddS4-AL remains open.

# 7 Experiments

In this section, we propose two sets of experiments [*] to demonstrate the practical relevance of our new revenue function and the corresponding algorithm. In our first set of experiments, our goal is to show the usefulness of revenue maximisation as a solution to find hierarchies that are closer to the ground truth. We consider a planted model and demonstrate the alignment between AARI scores, a supervised metric of goodness for clustering, and our proposed revenue function. In our second set of experiments, we aim to show that the heuristic proposed in Section 6 to maximize the revenue performs well in practice. On real datasets, we compare our approach to two different state of the art approaches in comparison-based hierarchical clustering.

## 7.1 Planted Model

In this first set of experiments, we study the behaviour of the proposed revenues in a controlled setting. Hence, we generate data using a planted model for comparison-based hierarchical clustering (Ghoshdastidar et al., 2019) and we use 3 triplets-based and 2 quadruplets-based methods to learn dendrograms.

**Data.** To generate the data in this first set of experiments, we use a standard planted model in comparison-based hierarchical clustering (Balakrishnan et al., 2011; Ghoshdastidar et al., 2019). Given $n$ objects, we create a real similarity matrix $S = (s_{ij})_{1 \leq i, j \leq n}$ such that $s_{ii} = 0$ and $s_{ij} = s_{ji}$ correspond to the similarity between objects $i$ and $j$. We assume that $(s_{ij})_{i<j}$ are independent Gaussians with $s_{ij} \sim \mathcal{N}(\mu_{ij}, \sigma^2)$. The choice of $\mu_{ij}$ defines a planted hierarchy, a complete binary tree of height $L$, built on top of $2^L$ ground clusters, that is sets of $n_0$ objects denoted as $\mathcal{C}_1, \mathcal{C}_2, ..., \mathcal{C}_{2^L}$. The total number of points (leaves) in the complete hierarchy is thus $n = n_0 2^L$. For every pair of objects $i, j$ that belong to the same ground cluster, $\mu_{ij} = \mu$—a constant. On the other hand, for objects $i, j$ from two distinct clusters, if $H(i \vee j)$ is rooted at level $\ell$, we define $\mu_{ij} = \mu - (L - \ell)\delta$. We observe that the tree is rooted at level-0 and the constants—separation $\delta$ and noise level $\sigma$—control the hardness of the problem. In particular, smaller values of $\delta$ make the similarities between examples that belong to the same cluster more difficult to distinguish from similarities between examples that belong to different clusters. The signal-to-noise ratio is thus $\frac{\delta}{\sigma}$. In all the experiments, we set $\mu = 0.8$, $\sigma = 0.1$, $n_0 = 30$, $L = 3$ and we vary $\delta \in \{0.02, 0.04, ..., 0.2\}$. Since we are in a comparison-based setting, we do not directly use the similarities of the planted model to learn dendrograms but instead generate comparisons. Given $\mathcal{T}_{all}$ and $\mathcal{Q}_{all}$ the sets containing all possible triplets and quadruplets (see preliminaries), we obtain $\mathcal{T} \subseteq \mathcal{T}_{all}$ and $\mathcal{Q} \subseteq \mathcal{Q}_{all}$ by uniformly sampling $kn^2$ comparisons with $k > 0$.

**Evaluation Function.** To measure the closeness between the dendrograms obtained by the different approaches and the ground truth trees, we use the Averaged Adjusted Rand Index (Ghoshdastidar et al., 2019). The AARI is an extension to hierarchies of a well-known measure in standard clustering called Adjusted Rand Index (ARI; see Hubert and Arabie, 1985). The underlying idea is to average the ARI obtained over the top $L$ levels of the tree. This measure takes values in $[0, 1]$ with higher values for more similar hierarchies, an AARI of 1 implying identical trees. Our goal is to empirically verify that the hierarchies with higher revenues are the ones closest to the ground truths as indicated by a higher AARI. Indeed, this would show that our revenue function is appropriate to evaluate the goodness of a dendrogram and that maximizing the revenue is indeed a good unsupervised way to select hierarchies. The results reported are averaged over 10 independent trials. [*] We defer the standard deviations to the appendix for the sake of readability.

**Methods.** We compare AddS3-AL and AddS4-AL, the two methods proposed in this work, to various comparison-based algorithms for learning dendrograms, such as 4K-AL (Ghoshdastidar et al., 2019), a

---

[*]The code is available at `https://github.com/jitaishik/Revenue_ComparisonHC.git`

[*]The randomness stems from three sources: the noise in the similarities, triplets selection, and the optimization procedure in tSTE. We fix the seeds to 0-9 in the 10 runs.

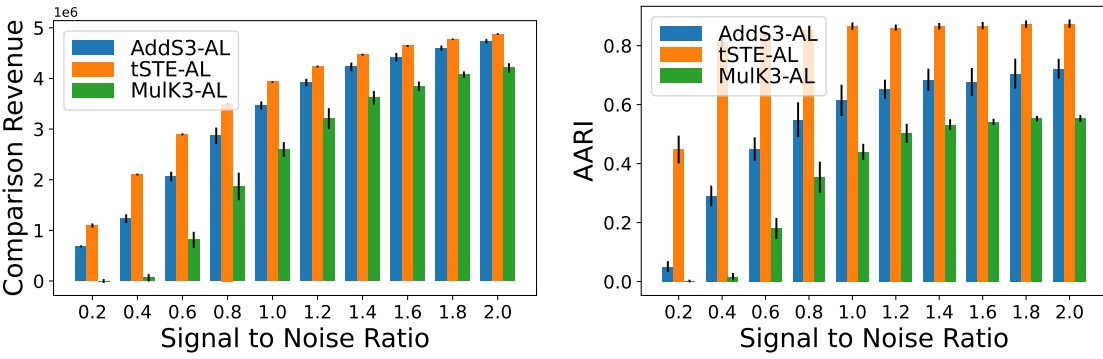

Figure 1: Revenue and AARI (higher is better) of several triplets-based methods using $n^2$ comparisons. Given various signal to noise ratios, a higher revenue implies higher AARI values (better dendrograms).

Table 1: Revenue and AARI of various methods for a fixed signal to noise ratio $\frac{\delta}{\sigma} = 1.5$ and varying number of triplets (decreased by factor of 2). The planted setting consists of a total of $n = 240$ objects. In each line the highest revenue and AARI are underlined, taking into account standard deviation (see appendix). This shows that the two measures are well aligned.

| Number of triplets | AddS3-AL | | tSTE-AL | | MulK3-AL | |
|---|---|---|---|---|---|---|
| | Revenue | AARI | Revenue | AARI | Revenue | AARI |
| $16n^2$ | $\underline{7.347 \times 10^7}$ | $\underline{0.937}$ | $7.300 \times 10^7$ | $0.877$ | $7.315 \times 10^7$ | $0.861$ |
| $8n^2$ | $\underline{3.667 \times 10^7}$ | $\underline{0.905}$ | $3.656 \times 10^7$ | $0.877$ | $3.636 \times 10^6$ | $0.855$ |
| $4n^2$ | $\underline{1.823 \times 10^7}$ | $\underline{0.862}$ | $1.825 \times 10^7$ | $\underline{0.874}$ | $1.795 \times 10^7$ | $0.830$ |
| $2n^2$ | $8.962 \times 10^6$ | $0.782$ | $\underline{9.130 \times 10^6}$ | $\underline{0.867}$ | $8.444 \times 10^6$ | $0.677$ |
| $n^2$ | $4.315 \times 10^6$ | $0.682$ | $\underline{4.559 \times 10^6}$ | $\underline{0.868}$ | $3.728 \times 10^6$ | $0.540$ |
| $n^2/2$ | $2.038 \times 10^6$ | $0.593$ | $\underline{2.277 \times 10^6}$ | $\underline{0.860}$ | $1.220 \times 10^6$ | $0.347$ |
| $n^2/4$ | $9.268 \times 10^5$ | $0.498$ | $\underline{1.137 \times 10^6}$ | $\underline{0.851}$ | $1.531 \times 10^5$ | $0.077$ |
| $n^2/8$ | $4.261 \times 10^5$ | $0.396$ | $\underline{5.728 \times 10^5}$ | $\underline{0.840}$ | $1.856 \times 10^4$ | $0.011$ |
| $n^2/16$ | $2.015 \times 10^5$ | $0.295$ | $\underline{2.858 \times 10^5}$ | $\underline{0.720}$ | $4.026 \times 10^3$ | $0.005$ |
| $n^2/32$ | $1.096 \times 10^5$ | $0.192$ | $\underline{1.450 \times 10^5}$ | $\underline{0.549}$ | $2.015 \times 10^3$ | $0.003$ |

quadruplets-based method, along with two triplets-based approaches MulK3-AL (Kleindessner and von Luxburg, 2017; Perrot et al., 2020) and tSTE-AL (Van Der Maaten and Weinberger, 2012). The former two are similarity-based approaches where the idea is use the comparisons to learn a similarity. The latter is an ordinal embedding approach where the idea is to recover a representation of the data that respects the comparisons as well as possible, and then use the cosine similarity $s_{ij} = \frac{\langle x_i, x_j \rangle}{||x_i||_2 ||x_j||_2}$ to compare the examples. To learn the dendrograms we then apply standard average linkage to the various similarities.

**Results.**  In Figure 1, we present the AARI and Revenue of different triplet-based methods for several signal to noise ratios using $n^2$ comparisons.[*] We observe that, given a set signal to noise ratio, the ordering between the methods remains the same for the revenue and the AARI, that is the method with the highest revenue is also the one with the highest AARI. In other words, a higher revenue indicates that the corresponding dendrogram is better. In Table 1, we verify that this remains true for a constant signal to noise ratio of 1.5 and various number of observed comparisons. In particular, we notice that when the revenue of AddS3-AL becomes higher than the revenue of tSTE-AL, that is using more than $4n^2$ triplets, the AARI also follows the same trend, thus confirming that selecting the dendrogram with the highest revenue is indeed a good

---

[*]Note that we also considered other amounts of comparisons. However, the trends were similar to the ones observed here and thus we chose to defer these results to the appendix.

Table 2: Experiments on real datasets. For the triplets-based methods, AddS3-AL tends to obtain the dendrograms with the best revenues. For the quadruplets-based approaches, AddS4-AL and 4K-AL obtain comparable results. Using the original Cosine similarities only yields slightly better hierarchies than the comparison-based methods. For first 3 datasets, multiple runs are used and standard deviation (see appendix) is considered for highlighting the best method(s).

| Dataset | Triplet | | | | Quadruplet | | |
|---|---|---|---|---|---|---|---|
| | AddS3-AL | tSTE-AL | MulK3-AL | Cosine-AL | AddS4-AL | 4K-AL | Cosine-AL |
| Zoo | $2.771{\times}10^5$ | $2.164{\times}10^5$ | $2.041{\times}10^5$ | $2.824{\times}10^5$ | $2.828{\times}10^5$ | $2.866{\times}10^5$ | $2.945{\times}10^5$ |
| Glass | $2.161{\times}10^6$ | $1.973{\times}10^6$ | $1.412{\times}10^6$ | $2.110{\times}10^6$ | $2.429{\times}10^6$ | $2.425{\times}10^6$ | $2.494{\times}10^6$ |
| MNIST | $1.893{\times}10^9$ | $2.061{\times}10^9$ | $1.724{\times}10^9$ | $2.064{\times}10^9$ | $1.905{\times}10^9$ | $1.884{\times}10^9$ | $2.075{\times}10^9$ |
| Car | $1.521{\times}10^5$ | $1.562{\times}10^5$ | $1.264{\times}10^5$ | - | $1.521{\times}10^5$ | $1.125{\times}10^5$ | - |
| Food | $6.137{\times}10^6$ | $5.993{\times}10^6$ | $6.096{\times}10^6$ | - | $6.137{\times}10^6$ | $6.137{\times}10^6$ | - |
| Vogue | $2.722{\times}10^4$ | $2.104{\times}10^4$ | $3.022{\times}10^3$ | - | $2.722{\times}10^4$ | $2.549{\times}10^4$ | - |
| Nature | $2.650{\times}10^5$ | $2.056{\times}10^5$ | $1.231{\times}10^5$ | - | $2.650{\times}10^5$ | $2.228{\times}10^5$ | - |
| Imagenet | $7.179{\times}10^7$ | $6.571{\times}10^7$ | $3.440{\times}10^7$ | - | $7.179{\times}10^7$ | $6.994{\times}10^7$ | - |

way to select meaningful hierarchies. In the appendix, we show that the same behaviour can be observed for various signal to noise ratios as well as in the quadruplet case.

We further investigate the dependence between AARI and the triplet revenue in Figure 2, where we plot the AARI and the corresponding triplet revenue (in log scale) for different runs and number of triplets, considered in Table 1. Although the variation of the revenue for increasing AARI seems to depend on the method under consideration, all plots show a monotonic trend. To validate this we use two measures of rank correlation—Kendall's-$\tau$ and Spearman's-$\rho$—that capture how well the dependence is captured as monotonic function. For both AddS3-AL and MulK3-AL, the rank correlations are greater than 0.9, indicating a highly monotonic trend. Although the rank correlations are smaller for tSTE-AL, but still large enough and corresponding $p$-value is about $10^{-17}$, indicating rank correlation.

## 7.2 Real Data

The previous experiments establish that our revenue functions are good at identifying meaningful dendrograms in an unsupervised way. In the following experiments, we investigate the behaviour of the proposed approaches on real data. In particular, we show that they are competitive with standard comparison-based hierarchical clustering approaches on various datasets.

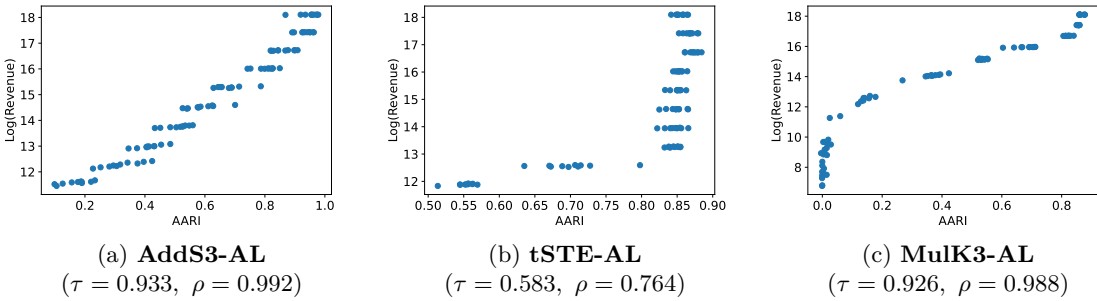

(a) **AddS3-AL**
($\tau = 0.933,\ \rho = 0.992$)

(b) **tSTE-AL**
($\tau = 0.583,\ \rho = 0.764$)

(c) **MulK3-AL**
($\tau = 0.926,\ \rho = 0.988$)

Figure 2: Scatter plots for the AARI and triplet revenue from different runs and varying number of triplets in the setting of Table 1. The three methods (AddS3, tSTE and MulK3) show different trends. However, in each case, the triplet revenue and AARI have high Kendall-$\tau$ and Spearman-$\rho$ correlation.

**Data.** We consider 8 different datasets. On the one hand, we consider 3 standard clustering datasets: Zoo, Glass, and MNIST (Heller and Ghahramani, 2005; LeCun et al., 2010; Vikram and Dasgupta, 2016). The Zoo dataset originally consisted of 101 animals each with 16 features. But we choose to remove the entry with class "girl" since we do not feel it belongs to the zoo dataset. The Glass dataset has 9 features for 214 examples. For the MNIST dataset we consider two subsets of the MNIST test dataset that originally contains 10000 examples distributed among the ten digits. A 2-dimensional embedding of the entire MNIST test data was constructed with t-SNE (van der Maaten, 2014). From this we randomly sampled 200 examples for each digit to form a dataset of 2000 entries and normalized the embeddings so that each example lies in $[-1, 1]$. Since we are in a comparison-based setting, we generate $n^2$ comparisons using the cosine similarity. To model mistakes from human annotators, we randomly and uniformly flip 5% of the comparisons (Emamjomeh-Zadeh and Kempe, 2018), where by flipping $(i, j, k)$ we mean replacing it with $(i, k, j)$. On the other hand, we consider 5 comparison-based datasets, Car, Food, Vogue Cover, Nature Scene and ImageNet Images v0.1, from the cblearn repository.[*] The number of objects and the kind of query used to obtain comparisons are summarized in Table 3. The comparisons are transformed into triplets (final number of triplets noted in Table 3), which are also used in the quadruplet setting. In Table 3, a central triplet is a query of the form—which of the three objects (i,j,k) is most central. Provided that the answer is i, this implies object i is more similar to both j and k than they are to each other. Two standard triplets (j,i,k) and (k,i,j) are thus obtained. An Odd-out triplet is a query of the form—which of the three objects (i,j,k) is the odd one out. If i is picked as the odd one, it gives two standard triplets of the form (j,k,i) and (k,j,i). A rank 2 from 8 query is of the form—among 8 objects $(i_0, \ldots, i_7)$, rank the 2 that appear to be most similar to the reference object $i_0$. If $i_1$ and $i_2$ are ranked as the most similar to $i_0$ in this order, then 11 standard triplets of the form $(i_0, i_1, i_k)_{k=2}^{7}$ and $(i_0, i_2, i_k)_{k=3}^{7}$ are obtained.

Table 3: Description of datasets used in the experiments.

| Dataset | Query | #Objects | #Triplets |
|---------|-------|----------|-----------|
| Zoo | Cosine Similarity | 100 | 100000 |
| Glass | Cosine Similarity | 214 | 45796 |
| MNIST | Cosine Similarity | 2000 | 4000000 |
| Car | Most Central Triplet | 60 | 14194 |
| Food | Standard Triplet | 100 | 190376 |
| Vogue | Odd-out Triplet | 60 | 2214 |
| Nature | Odd-out Triplet | 120 | 6710 |
| Imagenet | Rank 2 from 8 | 1000 | 328549 |

**Evaluation Function.** Since the datasets considered here do not come with a ground truth hierarchy, we cannot compute the AARI. Hence, we only report the revenue. The results reported are averaged over 10 independent trials[*] and defer the standard deviations to the appendix.

**Methods.** Besides the methods already used in the planted setting, we also consider the Cosine baseline where it is assumed that the pairwise cosine similarities are available, and we apply average linkage directly on the similarities used to generate the comparisons. This baseline is not applicable to the comparison-based datasets where we only have access to the comparisons and not to the similarities.

**Results.** The results are reported in Table 2. We can notice that AddS3-AL tends to be better than tSTE-AL and MulK3-AL while AddS4-AL and 4K-AL are comparable. As is expected, the Cosine baseline based on the original similarities obtains the best performances in most cases, but it only seems to yield slightly better hierarchies than the comparison-based methods. This would tend to confirm that hierarchical clustering with average linkage is indeed a problem that can be solved using only a limited number of comparisons, instead of using all similarities.

---

[*] https://github.com/dekuenstle/cblearn

[*] The randomness stems from two main sources: the triplets generation (in the Zoo, Glass, and MNIST dataset), the optimization procedure in tSTE (initialization, batch selection). We fix the seeds to 0-9 for the 10 runs. For all the other datasets and methods, every step is deterministic and, thus, we only need to report the results of a single run.

## 8    Conclusion

In this paper, we proposed novel revenue functions that allow us to measure the goodness of a dendrogram in an unsupervised way using only triplet or quadruplet comparisons. This suggest natural algorithms for hierarchical clustering based on the maximization of such revenues. Drawing theoretical connections with existing work on cost and revenue functions in standard hierarchical clustering, we propose two algorithms based on average linkage for hierarchical clustering using only comparisons. We empirically show that our revenue functions successfully identify the dendrograms that are closest to the ground truth. We also show that the proposed approaches to learn hierarchies perform well on real datasets and are competitive with state of the art methods.

We further used the proposed revenue function to resolve an open theoretical problem of recovering a latent hierarchy using fewer than $\Omega(n^3)$ passive triplets. We showed that $O(n^2 \log n/\epsilon^2)$ passive triplets suffice to obtain a $(1-\epsilon)$-approximation of the optimal triplet revenue. We conclude with the following open questions:
(i) Are $\Omega(n^2 \log n)$ passive triplets necessary for a $(1 - \epsilon)$-approximation?
(ii) At this point, we are unable to obtain a polynomial-time approximation scheme (PTAS) for the revenue maximisation problem, but we believe this should be possible. It may also be possible to obtain more efficient algorithms that are linear in time with respect to the number of triplets. A linear-time method exists for maximum quartet consistency (Snir and Yuster, 2011).

**Acknowledgments**

The work of A. Mandal was supported by the German Academic Exchange Service (DAAD) through the Working Internships in Science and Engineering (WISE) scholarship.The work of D. Ghoshdastidar is partly supported by the German Research Foundation through the DFG-ANR PRCI "ASCAI" (GH 257/3-1).

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

## A  Proof of Theorem 1

Recall the formulation of $Trev$ for a tree $H$ and a set of triplets $\mathcal{T}$:

$$
\begin{aligned}
Trev(H, \mathcal{T}) &= \sum_{i,j,k} \mathbb{I}_{(i,j,k)\in\mathcal{T}} \Big( |H(i \vee k)| - |H(i \vee j)| \Big) && \text{(sum over all distinct } i,j,k.\text{)} \\
&= \sum_{i,j,k} \mathbb{I}_{(i,j,k)\in\mathcal{T}} |H(i \vee k)| - \sum_{i,j,k} \mathbb{I}_{(i,j,k)\in\mathcal{T}} |H(i \vee j)| \\
&= \sum_{i,j,k} \mathbb{I}_{(i,k,j)\in\mathcal{T}} |H(i \vee j)| - \sum_{i,j,k} \mathbb{I}_{(i,j,k)\in\mathcal{T}} |H(i \vee j)| \\
&&& \text{(by change of variable between } j \text{ and } k \text{ in the first term.)} \\
&= \sum_{i\neq j} \Big( \sum_{k\neq i,j} \mathbb{I}_{(i,k,j)\in\mathcal{T}} - \mathbb{I}_{(i,j,k)\in\mathcal{T}} \Big) |H(i \vee j)| \\
&= \sum_{i<j} |H(i \vee j)| \times \Big( \sum_{k\neq i,j} \mathbb{I}_{(i,k,j)\in\mathcal{T}} - \mathbb{I}_{(i,j,k)\in\mathcal{T}} + \mathbb{I}_{(j,k,i)\in\mathcal{T}} - \mathbb{I}_{(j,i,k)\in\mathcal{T}} \Big) \\
&&& \text{(by change of variable between } i \text{ and } j \text{ when } i > j.\text{)} \\
&= \sum_{i<j} -s_{ij}^{AddS3} |H(i \vee j)|
\end{aligned}
$$

by definition of $s^{AddS3}$. This concludes the proof for the triplets-based revenue.

Using a similar approach, recall the formulation of $Qrev$ for a tree $H$ and quadruplet set $\mathcal{Q}$:

$$
\begin{aligned}
Qrev(H, \mathcal{Q}) &= \sum_{i,j,k,l} \mathbb{I}_{(i,j,k,l)\in\mathcal{Q}} \Big( |H(k \vee l)| - |H(i \vee j)| \Big) \\
&= \sum_{i,j,k,l} \mathbb{I}_{(i,j,k,l)\in\mathcal{Q}} |H(k \vee l)| - \sum_{i,j,k,l} \mathbb{I}_{(i,j,k,l)\in\mathcal{Q}} |H(i \vee j)| \\
&&& \text{(sum over all } i < j, k < l, (i,j) \neq (k,l).\text{)} \\
&= \sum_{i,j,k,l} \mathbb{I}_{(k,l,i,j)\in\mathcal{Q}} |H(i \vee j)| - \sum_{i,j,k,l} \mathbb{I}_{(i,j,k,l)\in\mathcal{Q}} |H(i \vee j)| \\
&&& \text{(by swapping the role of } i,j \text{ and } k,l \text{ in the first term.)} \\
&= \sum_{i<j} \Big( \sum_{\substack{k<l \\ (k,l)\neq(i,j)}} \mathbb{I}_{(k,l,i,j)\in\mathcal{Q}} - \mathbb{I}_{(i,j,k,l)\in\mathcal{Q}} \Big) |H(i \vee j)| \\
&= \sum_{i<j} -s_{ij}^{AddS4} |H(i \vee j)|
\end{aligned}
$$

by definition of $s^{AddS4}$. This concludes the proof for the quadruplets-based revenue.

## B  Proof of Proposition 2

The proposition follows immediately from the following two lemmas.

**Lemma 5.** *Let $H_0$ be a binary tree on $[n]$ and $\mathcal{T}_0$ be the set of triplets induced by $H_0$. Then*

$$
\mathcal{T}_0 = \arg\max_{\mathcal{T}} Trev(H_0, \mathcal{T}), \tag{7}
$$

*where the maximum is unique over all triplet sets that are induced by some binary tree on $[n]$.*

**Lemma 6.** *Let $H_0, H_1$ be two binary trees on $[n]$ and $\mathcal{T}_0, \mathcal{T}_1$ be the set of triplets induced by $H_0$ and $H_1$, respectively. Then $Trev(H_0, \mathcal{T}_1) = Trev(H_1, \mathcal{T}_0)$.*

Combining the above two lemmas, we obtain that

$$Trev(H_0, \mathcal{T}_0) > Trev(H_0, \mathcal{T}_1) = Trev(H_1, \mathcal{T}_0)$$

for any tree $H_1$ and corresponding set of triplets $\mathcal{T}_1$. This directly implies the Proposition 2. We now complete the proof by proving Lemmas 5 and 6.

*Proof of Lemma 5.* Recall that

$$Trev(H_0, \mathcal{T}_0) = \sum_{(i,j,k) \in \mathcal{T}_0} \underbrace{\left( |H_0(i \vee k)| - |H_0(i \vee j)| \right)}_{=:D_0(i,j,k)}$$

is a sum of positive terms. For convenience, we denote each difference by $D_0(i,j,k) > 0$.

Let $\mathcal{T}_1$ be the set of triplets generated by another tree $H_1$. Note that at least one pair of triplets in $\mathcal{T}_1$ has to be different from $\mathcal{T}_0$, otherwise $H_1$ and $H_0$ would be isomorphic transformations of one another. Without loss of generality, assume that the pair $(i,j,k), (j,i,k)$ has been replaced by the pair $(i,k,j), (k,i,j)$, that is, $i,k$ are merged in $H_1$ before they are merged to $j$. Observe that we can write

$$Trev(H_0, \mathcal{T}_0) - Trev(H_0, \mathcal{T}_1) = \sum_{\substack{(i,j,k), \\ (i,k,j) \\ \in \mathcal{T}_0 \setminus \mathcal{T}_1}} D_0(i,j,k) + D_0(j,i,k) - D_0(i,k,j) - D_0(k,i,j)$$

where each term in the summation can be computed as

$$
\begin{aligned}
&D_0(i,j,k) + D_0(j,i,k) - D_0(i,k,j) - D_0(k,i,j) \\
&= |H_0(i \vee k)| + |H_0(j \vee k)| - 2|H_0(i \vee j)| - \left( |H_0(i \vee j)| + |H_0(k \vee j)| - 2|H_0(i \vee k)| \right) \\
&= 3D_0(i,j,k),
\end{aligned}
$$

which is strictly positive for every $(i,j,k) \in \mathcal{T}_0$. Summing over all $(i,j,k), (j,i,k) \in \mathcal{T}_0 \setminus \mathcal{T}_1$, we have that $Trev(H_0, \mathcal{T}_0) > Trev(H_0, \mathcal{T}_1)$ for any $\mathcal{T}_1$ generated by another tree $H_1$. □

*Proof of Lemma 6.* Let $\{s_{0ij}\}_{i,j}$ be the pairwise AddS3 similarity induced by $\mathcal{T}_0$, and $\{s_{1ij}\}_{i,j}$ be the AddS3 similarity from $\mathcal{T}_1$. Due to the definition of $\mathcal{T}_0$, we note that, for any $k \neq i, j$, the term $(\mathbb{I}_{(i,j,k) \in \mathcal{T}_0} - \mathbb{I}_{(i,k,j) \in \mathcal{T}_0} + \mathbb{I}_{(j,i,k) \in \mathcal{T}_0} - \mathbb{I}_{(j,k,i) \in \mathcal{T}_0})$ either takes the value 2 if $k \notin (i \vee j)$— that is, $i,j$ is merged in $H_0$ before $k$—or the value $-1$ if $k$ is merged to either $i$ or $j$ before $(i \vee j)$. Summing over all $k \neq i, j$ gives

$$s_{0ij} = 2(n - |H_0(i \vee j)|) - (|H_0(i \vee j)| - 2) = 2n + 2 - 3|H_0(i \vee j)|$$

for every $i, j$. Using the same arguments $s_{1ij}$ can be expressed as $s_{1ij} = 2n + 2 - 3|H_1(i \vee j)|$. We can now use the equivalence in Theorem 1 to write

$$
\begin{aligned}
Trev(H_0, \mathcal{T}_1) - Trev(H_1, \mathcal{T}_0) &= -\sum_{i<j}(s_{1ij}|H_0(i \vee j)| - s_{0ij}|H_1(i \vee j)|) \\
&= -\sum_{i<j}\left( (2n + 2 - 3|H_1(i \vee j)|)|H_0(i \vee j)| - (2n + 2 - 3|H_0(i \vee j)|)|H_1(i \vee j)| \right) \\
&= -(2n+2)\sum_{i<j}\left( |H_0(i \vee j)| - |H_1(i \vee j)| \right),
\end{aligned}
$$

which is zero, since $\sum_{i<j} |H_0(i \vee j)| = \sum_{i<j} |H_1(i \vee j)| = \frac{1}{3}(n^3 - n)$ is Dasgupta's cost for any tree on $[n]$ when all pairwise similarities are 1 (Dasgupta, 2016, Theorem 3). Hence, the claim. □

## C  Proof of Theorem 3 and Corollary 4

We first state and prove two lemmas that are essential for the proof of Theorem 3. The first lemma shows that $Trev(H_0, \mathcal{T}_0) = \Omega(n^4)$. The second lemma derives concentration inequalities for the AddS3 similarities $s_{ij}$, which is then used in the proof of Theorem 3 to derive bound on $|Trev(H, \mathcal{T}) - Trev(H, \mathcal{T}_0)|$ for all $H$, and subsequently arrive at the claim.

**Lemma 7.** *Given $\epsilon \in (0, 1)$ and $n \geq 8/\epsilon$, the following holds. If $H_0$ is a hierarchy on $[n]$ and $\mathcal{T}_0$ is the set of triplets induced by $H_0$,*

$$Trev(H_0, \mathcal{T}_0) \geq \frac{(1 - \epsilon)n^4}{12}.$$

*Proof.* We start with the equivalence in Theorem 1 to write the revenue as $Trev(H_0, \mathcal{T}_0) = -\sum_{i<j} s_{0ij} |H_0(i \vee j)|$, where $s_{0ij}$ is the pairwise AddS3 similarity induced by $\mathcal{T}_0$. Due to the definition of $\mathcal{T}_0$, we note that, for any $k \neq i, j$, the term $(\mathbb{I}_{(i,j,k) \in \mathcal{T}_0} - \mathbb{I}_{(i,k,j) \in \mathcal{T}_0} + \mathbb{I}_{(j,i,k) \in \mathcal{T}_0} - \mathbb{I}_{(j,k,i) \in \mathcal{T}_0})$ either takes the value 2 if $k \notin (i \vee j)$—that is, $i, j$ is merged in $H_0$ before $k$—or the value $-1$ if $k$ is merged to either $i$ or $j$ before $(i \vee j)$. Summing over all $k \neq i, j$ gives

$$s_{0ij} = 2(n - |H_0(i \vee j)|) - (|H_0(i \vee j)| - 2) = 2n + 2 - 3|H_0(i \vee j)|$$

for every $i, j$. Let $N = (i \vee j)$ denote the least common ancestor of $i, j$ in $H_0$, and $N_1, N_2$ be the two children of $N$. Note that $|N_1| \cdot |N_2|$ pairs of $i, j$ are merged at $N$. Hence, we can rewrite the revenue as

$$Trev(H_0, \mathcal{T}_0) = -\sum_{i<j} s_{0ij} |H_0(i \vee j)| = \sum_{i<j} |H_0(i \vee j)| (3|H_0(i \vee j)| - 2n - 2)$$

$$= \sum_{N \in H_0} |N_1| \cdot |N_2| \cdot |N| \cdot (3|N| - 2n - 2)$$

$$= 3 \sum_{N \in H_0} |N_1||N_2||N|^2 - (2n + 2) \sum_{N \in H_0} |N_1||N_2||N|$$

where the summations are over all internal nodes $N$ in the tree $H_0$, with $N_1, N_2$ deenoting the two children of $N$. The second summation is Dasgupta's cost for any tree on $[n]$ with all pairwise similarities as 1, and evaluates to $\frac{n^3 - n}{3}$ (Dasgupta, 2016, Theorem 3). On the other hand, we claim that the first sum has lower bound $\sum_{N \in H_0} |N_1||N_2||N|^2 \geq \frac{n^4}{4}$.

We prove this claim through induction on $n$. The claim is easy to verify for $n = 2, 3$. For $n \geq 4$, we assume that claim holds for any $H_0$ with $k$ leaves, when $k < n$ (equivalently, $H_0$ on $[k]$). Consider the tree $H_0$ on $[n]$ such that the root node is split into two nodes of size $n_1, n_2 < n$ (note $n_1 + n_2 = n$). From our inductive hypothesis,

$$\sum_{N \in H_0} |N_1||N_2||N|^2 \geq n_1 n_2 n^2 + \frac{n_1^4}{4} + \frac{n_2^4}{4}$$

$$= \frac{1}{4} \left(4n_1^3 n_2 + 8n_1^2 n_2^2 + 4n_1 n_2^3 + n_1^4 + n_2^4\right)$$

$$\geq \frac{1}{4}(n_1 + n_2)^4 = \frac{n^4}{4}$$

which proves the claim for any $n$. Combining all terms, we have

$$Trev(H_0, \mathcal{T}_0) \geq \frac{3n^4}{4} - \frac{(2n + 2)(n^3 - n)}{3} = \frac{n^4}{12} - \frac{2}{3}(n^3 - n^2 - n).$$

Now, given $\epsilon \in (0, 1)$, notice that $\frac{2}{3}(n^3 - n^2 - n) \leq \frac{\epsilon n^4}{12}$ for any $n > \frac{8}{\epsilon}$, which proves the lemma. $\square$

We now state and prove the concentration results for the AddS3 similarity computed from the sampled triplet set $\mathcal{T}$. We first recall the sampling and introduce some notations. For any $n$, with probability $p_n \in (0, 1)$, a pair of triplets $(i, j, k), (j, i, k) \in \mathcal{T}_0$ is included in $\mathcal{T}$, independent of other pairs. To formalise this, we define the random variable $\chi_{ijk} = \chi_{jik} \sim \text{Bernoulli}(p_n)$ such that the collection $\{\chi_{ijk} : i < j, k \neq i, j\}$ are mutually independent. If $s_{ij}$ denotes the pairwise AddS3 similarity, computed using $\mathcal{T}$, then observe that

$$s_{ij} = \sum_{k \neq i, j} \chi_{ijk}(\mathbb{I}_{(i,j,k) \in \mathcal{T}_0} - \mathbb{I}_{(i,k,j) \in \mathcal{T}_0} + \mathbb{I}_{(j,i,k) \in \mathcal{T}_0} - \mathbb{I}_{(j,k,i) \in \mathcal{T}_0}) \tag{8}$$

Hence, for a fixed $H_0$—and $\mathcal{T}_0$—the similaritiy $s_{ij}$ is a weighted sum of independent Bernoullis, with weights either 2 or $-1$ (cf. proof of Lemma 7). As a consequence, $\mathbb{E}[s_{ij}] = p_n s_{0ij}$, where the expectation is with respect to sampling, and furthermore we can state the following concentration for all pairwise similarities.

**Lemma 8.** *Assume $n \geq 8$. Let $\mathcal{T}$ denote a random subset of $\mathcal{T}_0$ (obtained from the aforementioned sampling), and $\{s_{ij}\}_{i<j}, \{s_{0ij}\}_{i<j}$ denote the pairwise AddS3 similarities computed using triplets in $\mathcal{T}$ and $\mathcal{T}_0$, respectively. For any $\alpha > 0$, if $p_n > (\alpha + 2) \log n / n$, then with probability $1 - 2n^{-\alpha}$,*

$$\frac{p_n n^3}{10} \leq |\mathcal{T}| \leq \frac{p_n n^3}{2} \quad and \quad \max_{i<j} |s_{ij} - p_n s_{0ij}| \leq 4\sqrt{(\alpha + 2) p_n n \log n}.$$

*Proof.* We first derive the bound on $\max_{i<j} |s_{ij} - p_n s_{0ij}|$. From the expression of $s_{ij}$, mentioned above, we note that $s_{ij} - p_n s_{0ij}$ is a sum of $(n-2)$ independent mean zero random variables, with each term in $[-2, 2]$ and variance bounded by $4p_n$. By Bernstein's inequality,

$$\mathbb{P}(|s_{ij} - p_n s_{0ij}| > \delta) \leq 2 \exp\left(-\frac{\delta^2}{8p_n(n-2) + \frac{4}{3}\delta}\right).$$

For any $\alpha > 0$, if $p_n > (\alpha + 2) \log n / n$ and setting $\delta = 4\sqrt{(\alpha + 2) p_n n \log n}$, we have $|s_{ij} - p_n s_{0ij}| > 4\sqrt{(\alpha + 2) p_n n \log n}$ with probability $\leq 2n^{-(\alpha+2)}$. Using union bound over all $\binom{n}{2} < n^2/2$ pairs of $i, j$, we have that $\max_{i<j} |s_{ij} - p_n s_{0ij}|$ exceeds the claim bound with probability $\leq n^{-\alpha}$.

To derive the bound on $|\mathcal{T}|$, we first bound $|\mathcal{T}_0|$. From definition of $\mathcal{T}_0$, every internal node $N \in H_0$ contributes $|N_1||N_2|(|N| - 2)$ triplets to $\mathcal{T}_0$ since merger of every $i, j$ contributes to $|N| - 2$ triplets, one for each $k$ that is either merged with $i$ or $j$ at a lower level. Hence,

$$|\mathcal{T}_0| = \sum_{N \in H_0} |N_1||N_2|(|N| - 2) = \sum_{N \in H_0} |N_1||N_2||N| - 2 \sum_{N \in H_0} |N_1||N_2| = \frac{n^3 - n}{3} - 2 \cdot \frac{n^2}{2} = \frac{n^3 - 3n^2 - n}{3}.$$

The first summation follows from (Dasgupta, 2016, Theorem 3), whereas the second sum follows from induction with hypothesis that the sum evaluates to $n^2/2$. We now note that $\mathbb{E}[|\mathcal{T}|] = p_n |\mathcal{T}_0|$ and apply multiplicative Chernoff bound to get that $\frac{1}{2} p_n |\mathcal{T}_0| \leq |\mathcal{T}| \leq \frac{3}{2} p_n |\mathcal{T}_0|$ with probability $1 - 2e^{-p_n |\mathcal{T}_0|/12}$. To simplify the terms, we use $|\mathcal{T}_0| = \frac{n^3 - 3n^2 - n}{3} \in [\frac{n^3}{5}, \frac{n^3}{3}]$, where the lower bound holds for $n \geq 8$. This leads to the bounds on $|\mathcal{T}|$, whereas for the probability, note that $2e^{-p_n |\mathcal{T}_0|/12} \leq 2n^{-(\alpha+2)} \leq n^{-\alpha}$, where the first inequality follows using $|\mathcal{T}_0| \geq n^3/5, p_n > (\alpha + 2) \log n / n$ and $n \geq 8$. $\square$

Below, we prove Theorem 3 using Lemmas 7–8.

*Proof of Theorem 3.* We first derive bounds on the deviation of the revenue $Trev$ of any tree $H$ due to sampling. The concentration of $\{s_{ij}\}_{i<j}$ ensures that we can state a deviation bound that uniformly holds

for all $H$, as shown below. From the equivalence in Theorem 1, we write for any $H$,

$$
\begin{aligned}
|Trev(H, \mathcal{T}) - p_n Trev(H, \mathcal{T}_0)| &= \left| \sum_{i<j} (s_{ij} - p_n s_{0ij}) |H(i \vee j)| \right| \\
&\leq \sum_{i<j} |s_{ij} - p_n s_{0ij}| \cdot |H(i \vee j)| \\
&\leq 4\sqrt{(\alpha+2)p_n n \log n} \cdot \sum_{i<j} |H(i \vee j)|,
\end{aligned}
$$

where the last bound holds with probability $1 - n^{-\alpha}$ due to Lemma 8. Note that $\sum_{i<j} |H(i \vee j)|$ is Dasgupta's cost of tree $H$ on $[n]$ if all pairwise similarities are 1, and hence the summation is $\frac{n^3 - n}{3} < \frac{n^2}{3}$. We conclude that, with probability $1 - n^{-\alpha}$,

$$
\max_H |Trev(H, \mathcal{T}) - p_n Trev(H, \mathcal{T}_0)| < (4/3)\sqrt{(\alpha+2)p_n n^7 \log n}.
$$

We now write

$$
\begin{aligned}
Trev(\widehat{H}, \mathcal{T}_0) &\geq \frac{1}{p_n} Trev(\widehat{H}, \mathcal{T}) - \frac{4}{3}\sqrt{\frac{(\alpha+2)n^7 \log n}{p_n}} \\
&\geq \frac{1}{p_n} Trev(H_0, \mathcal{T}) - \frac{4}{3}\sqrt{\frac{(\alpha+2)n^7 \log n}{p_n}} \geq Trev(H_0, \mathcal{T}_0) - \frac{8}{3}\sqrt{\frac{(\alpha+2)n^7 \log n}{p_n}},
\end{aligned}
$$

where the first and third inequalities follow from the deviation bound stated above, and the second inequality holds since $\widehat{H}$ maximises $Trev(H, \mathcal{T})$. For $p_n > 2^{12}(\alpha+2)\log n / n\epsilon^2$, the second term is smaller than $\epsilon n^4/24 \leq \epsilon(1-\epsilon)n^4/12 \leq \epsilon\, Trev(H_0, \mathcal{T}_0)$, where the first inequality uses the fact $\epsilon \leq 1/2$ and the second inequality is due to Lemma 7. Hence the claim. □

*Proof of Corollary 4.* In the noisy setting, the random flipping can be modelled through the independent variables $\{\zeta_{jk}^i \ : \ i, j < k\}$ where $\zeta_{jk}^i \sim$ Bernoulli($\delta$) is the indicator for triplet $(i, j, k)$ to be flipped with triplet $(i, k, j)$. Using the notation of equation 8, the variable $\xi_{ijk}\zeta_{jk}^i$ indicates $(i, k, j) \in \mathcal{T}'$ (noisy triplet) whereas $\xi_{ijk}(1 - \zeta_{jk}^i)$ indicates correct triplet $(i, j, k) \in \mathcal{T}'$. Hence,

$$
s_{ij} = \sum_{k \neq i,j} \chi_{ijk}(1 - 2\zeta_{jk}^i)(\mathbb{I}_{(i,j,k)\in\mathcal{T}_0} - \mathbb{I}_{(i,k,j)\in\mathcal{T}_0}) + \chi_{ijk}(1 - 2\zeta_{ik}^j)(\mathbb{I}_{(j,i,k)\in\mathcal{T}_0} - \mathbb{I}_{(j,k,i)\in\mathcal{T}_0}), \tag{9}
$$

and $\mathbb{E}[s_{ij}] = (1 - 2\delta)p_n s_{0ij}$. Following the arguments of Lemma 8, we can that, with probability $1 - n^{-\alpha}$,

$$
\max_{i<j} |s_{ij} - (1 - 2\delta)p_n s_{0ij}| < 4\sqrt{(\alpha+2)p_n n \log n}
$$

where the deviation bound is same as in Lemma 8 since the same variance bound holds for the independent random terms in the summation in equation 9. Subsequently, following the proof of Theorem 3, we have with probability $1 - n^{-\alpha}$,

$$
\max_H |Trev(H, \mathcal{T}) - (1 - 2\delta)p_n \cdot Trev(H, \mathcal{T}_0)| = (4/3)\sqrt{(\alpha+2)p_n n^7 \log n}.
$$

and so

$$
\begin{aligned}
Trev(\widehat{H}, \mathcal{T}_0) &\geq \frac{1}{(1-2\delta)p_n} Trev(\widehat{H}, \mathcal{T}) - \frac{4}{3(1-2\delta)}\sqrt{\frac{(\alpha+2)n^7 \log n}{p_n}} \\
&\geq \frac{1}{(1-2\delta)p_n} Trev(H_0, \mathcal{T}) - \frac{4}{3(1-2\delta)}\sqrt{\frac{(\alpha+2)n^7 \log n}{p_n}} \geq Trev(H_0, \mathcal{T}_0) - \frac{8}{3(1-2\delta)}\sqrt{\frac{(\alpha+2)n^7 \log n}{p_n}}.
\end{aligned}
$$

Following the proof of Theorem 3, second term is $\leq \epsilon n^4/24$ for $p_n > \frac{2^{12} \cdot (\alpha+2)\log n}{n\epsilon^2(1-2\delta)^2}$. □

## D  Standard Deviation on Real Data

In Table 4, we provide the standard deviations for the real data experiments that were omitted in the main paper.

Table 4: Experiments on real datasets. For the triplets-based methods, AddS3-AL typically obtains dendrograms with the best revenues. For the quadruplets-based methods, AddS4-AL and 4K-AL show similar results. Using Cosine similarities yields slightly better hierarchies than comparison-based methods.

| Dataset | Triplet | | | |
|---|---|---|---|---|
| | AddS3-AL | tSTE-AL | MulK3-AL | Cosine-AL |
| Zoo | $\underline{2.77 \times 10^5 \pm 5 \times 10^3}$ | $2.16 \times 10^5 \pm 8 \times 10^3$ | $2.04 \times 10^5 \pm 2 \times 10^4$ | $\underline{2.82 \times 10^5 \pm 3 \times 10^3}$ |
| Glass | $\underline{2.16 \times 10^6 \pm 4 \times 10^4}$ | $1.97 \times 10^6 \pm 3 \times 10^4$ | $1.41 \times 10^6 \pm 5 \times 10^4$ | $\underline{2.11 \times 10^6 \pm 2 \times 10^4}$ |
| MNIST | $1.89 \times 10^9 \pm 4 \times 10^7$ | $\underline{2.06 \times 10^9 \pm 4 \times 10^7}$ | $1.72 \times 10^9 \pm 6 \times 10^7$ | $\underline{2.06 \times 10^9 \pm 2 \times 10^6}$ |
| Car | $1.52 \times 10^5$ | $\underline{1.56 \times 10^5 \pm 2 \times 10^3}$ | $1.26 \times 10^5$ | - |
| Food | $\underline{6.14 \times 10^6}$ | $5.99 \times 10^6 \pm 2 \times 10^4$ | $6.10 \times 10^6$ | - |
| Vogue | $\underline{2.72 \times 10^4}$ | $2.10 \times 10^4 \pm 1 \times 10^3$ | $3.02 \times 10^3$ | - |
| Nature | $\underline{2.65 \times 10^5}$ | $2.06 \times 10^5 \pm 8 \times 10^3$ | $1.23 \times 10^5$ | - |
| Imagenet | $\underline{7.18 \times 10^7}$ | $6.57 \times 10^7 \pm 8 \times 10^5$ | $3.44 \times 10^7$ | - |

| Dataset | Quadruplet | | |
|---|---|---|---|
| | AddS4-AL | 4K-AL | Cosine-AL |
| Zoo | $2.83 \times 10^5 \pm 1 \times 10^4$ | $2.87 \times 10^5 \pm 1 \times 10^4$ | $\underline{2.95 \times 10^5 \pm 3 \times 10^3}$ |
| Glass | $2.43 \times 10^6 \pm 3 \times 10^4$ | $2.43 \times 10^6 \pm 3 \times 10^4$ | $\underline{2.49 \times 10^6 \pm 1 \times 10^4}$ |
| MNIST | $1.91 \times 10^9 \pm 4 \times 10^7$ | $1.88 \times 10^9 \pm 3 \times 10^7$ | $\underline{2.08 \times 10^9 \pm 2 \times 10^6}$ |
| Car | $\underline{1.52 \times 10^5}$ | $1.13 \times 10^5$ | - |
| Food | $\underline{6.14 \times 10^6}$ | $6.14 \times 10^6$ | - |
| Vogue | $\underline{2.72 \times 10^4}$ | $2.55 \times 10^4$ | - |
| Nature | $\underline{2.65 \times 10^5}$ | $2.23 \times 10^5$ | - |
| Imagenet | $\underline{7.18 \times 10^7}$ | $6.99 \times 10^7$ | - |

## E  Additional Results on the Planted Model

In this section, we provide additional results on the Planted Model presented in Section 7.1 of the main paper. In Figure 3, we present the results obtained $n^2/2$, $n^2$ and $2n^2$ triplet comparisons respectively. Similarly, Figure 4 displays the results obtained using $n^2/2$, $n^2$ and $2n^2$ quadruplet comparisons respectively. In all these figures, we notice that, given a set signal to noise ratio, the ordering between the methods remains the same for the revenue and the AARI, that is the method with the highest revenue also has the highest AARI. In other words, a higher revenue indicates a better dendrogram.[*]

In Table 5 we verify that this remains true for constant signal to noise ratios of 1.5, and halving number of comparisons (Table 1 is an abbrieved version of Table 5). The highest revenue and AARI are underlined. We can notice that, when the revenue of AddS3-AL becomes higher than the revenue of tSTE-AL, the AARI also follows the same trend, thus confirming that selecting the dendrogram with the highest revenue is indeed a good way to select meaningful hierarchies.

## F  Results on the Planted Model with Noisy Comparisons

In the main paper, we only used the planted model to generate comparisons with no noise. In this section, we show that our findings remain true even when some of the comparisons are noisy, that is randomly

---

[*]In Figures 3–6, we use current time as the seeds for random numbers for each run.

flipped with a probability of 5%. In Figure 5, we present the results obtained using $n^2/2$, $n^2$ and $2n^2$ noisy triplet comparisons respectively. In Figure 6, we present the results obtained using $n^2/2$, $n^2$ and $2n^2$ noisy quadruplet comparisons respectively. We notice that, given a set signal to noise ratio, the ordering between the methods remains the same for the revenue and the AARI, that is the method with the highest revenue is also the one with the highest AARI. In other words, a higher revenue indicates a better dendrogram.

Table 5: Revenue and AARI of various methods for a signal to noise ratio of 1.5 and halving of comparisons. In each line the highest revenue and the highest AARI are underlined, showing that the two measures are well aligned.

| Number of triplets | AddS3-AL | | tSTE-AL | |
|---|---|---|---|---|
| | Revenue | AARI | Revenue | AARI |
| $16n^2$ | $7.347 \times 10^7 \pm 1.3 \times 10^5$ | $0.937 \pm 0.024$ | $7.300 \times 10^7 \pm 8.7 \times 10^4$ | $0.877 \pm 0.007$ |
| $8n^2$ | $3.667 \times 10^7 \pm 1.7 \times 10^5$ | $0.905 \pm 0.020$ | $3.656 \times 10^7 \pm 8.8 \times 10^4$ | $0.877 \pm 0.007$ |
| $4n^2$ | $1.823 \times 10^7 \pm 1.0 \times 10^5$ | $0.862 \pm 0.023$ | $1.825 \times 10^7 \pm 7.0 \times 10^4$ | $0.874 \pm 0.012$ |
| $2n^2$ | $8.962 \times 10^6 \pm 9.8 \times 10^4$ | $0.782 \pm 0.042$ | $9.130 \times 10^6 \pm 4.0 \times 10^4$ | $0.867 \pm 0.014$ |
| $n^2$ | $4.315 \times 10^6 \pm 9.7 \times 10^4$ | $0.682 \pm 0.047$ | $4.559 \times 10^6 \pm 2.0 \times 10^4$ | $0.868 \pm 0.012$ |
| $n^2/2$ | $2.038 \times 10^6 \pm 6.8 \times 10^4$ | $0.593 \pm 0.037$ | $2.277 \times 10^6 \pm 1.4 \times 10^4$ | $0.860 \pm 0.014$ |
| $n^2/4$ | $9.268 \times 10^5 \pm 4.2 \times 10^4$ | $0.498 \pm 0.035$ | $1.137 \times 10^6 \pm 1.0 \times 10^4$ | $0.851 \pm 0.065$ |
| $n^2/8$ | $4.261 \times 10^5 \pm 2.5 \times 10^4$ | $0.396 \pm 0.033$ | $5.728 \times 10^5 \pm 6.2 \times 10^3$ | $0.840 \pm 0.010$ |
| $n^2/16$ | $2.015 \times 10^5 \pm 1.2 \times 10^4$ | $0.295 \pm 0.041$ | $2.858 \times 10^5 \pm 3.3 \times 10^3$ | $0.720 \pm 0.057$ |
| $n^2/32$ | $1.096 \times 10^5 \pm 8.5 \times 10^3$ | $0.192 \pm 0.057$ | $1.450 \times 10^5 \pm 2.2 \times 10^3$ | $0.549 \pm 0.025$ |

| Number of triplets | MulK3-AL | |
|---|---|---|
| | Revenue | AARI |
| $16n^2$ | $7.315 \times 10^7 \pm 1.3 \times 10^5$ | $0.861 \pm 0.005$ |
| $8n^2$ | $3.636 \times 10^7 \pm 9.7 \times 10^4$ | $0.855 \pm 0.003$ |
| $4n^2$ | $1.795 \times 10^7 \pm 9.3 \times 10^4$ | $0.830 \pm 0.009$ |
| $2n^2$ | $8.444 \times 10^6 \pm 1.3 \times 10^5$ | $0.677 \pm 0.041$ |
| $n^2$ | $3.728 \times 10^6 \pm 1.4 \times 10^5$ | $0.540 \pm 0.016$ |
| $n^2/2$ | $1.220 \times 10^6 \pm 1.7 \times 10^5$ | $0.347 \pm 0.050$ |
| $n^2/4$ | $1.531 \times 10^5 \pm 8.9 \times 10^4$ | $0.077 \pm 0.047$ |
| $n^2/8$ | $1.856 \times 10^4 \pm 1.2 \times 10^4$ | $0.011 \pm 0.011$ |
| $n^2/16$ | $4.026 \times 10^3 \pm 8.0 \times 10^3$ | $0.005 \pm 0.004$ |
| $n^2/32$ | $2.015 \times 10^3 \pm 3.5 \times 10^3$ | $0.0003 \pm 0.001$ |

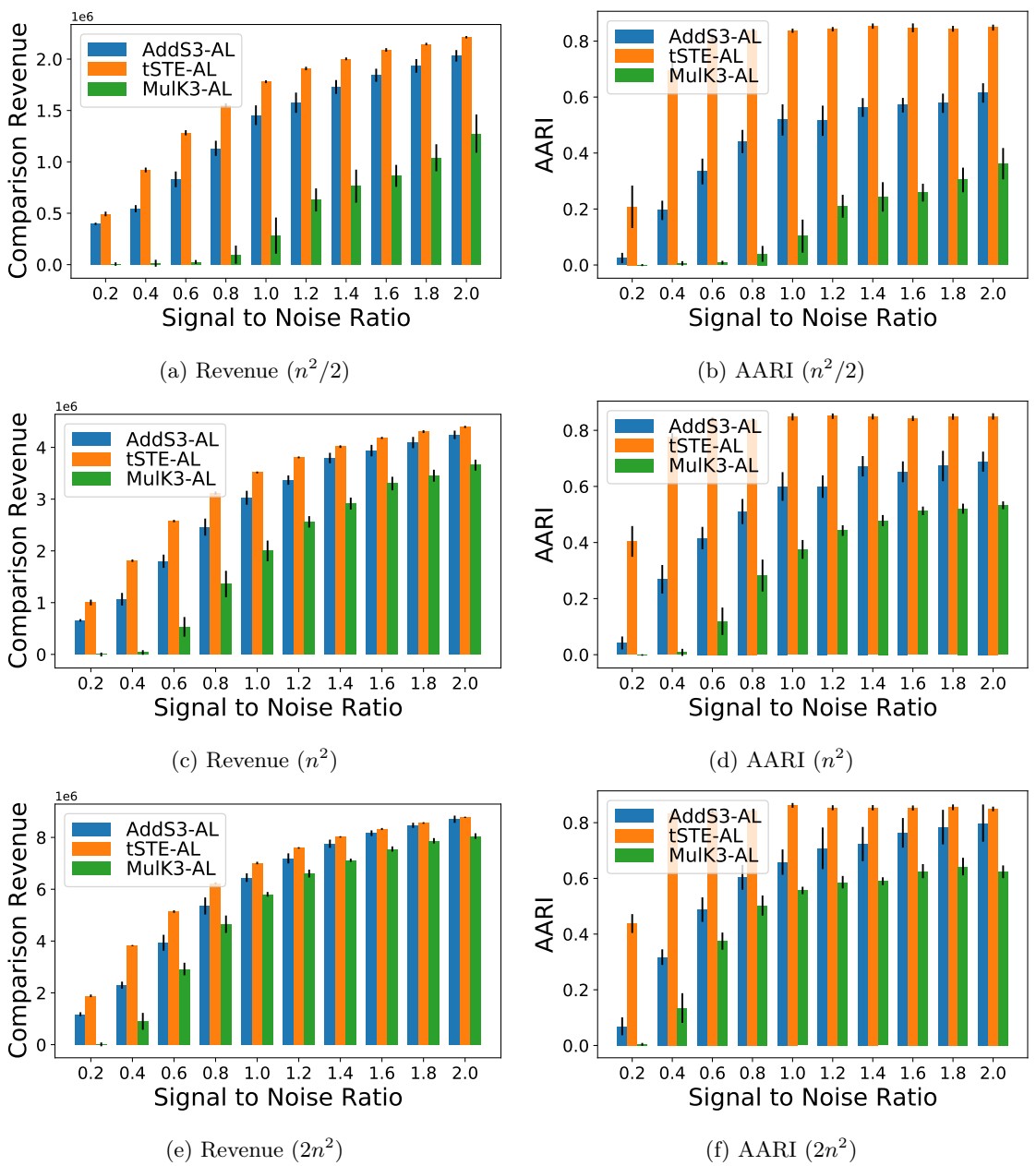

Figure 3: Revenue and AARI (higher is better) of several triplets-based methods using respectively $n^2/2$ comparisons (a-b), $n^2$ comparisons (c-d), and $2n^2$ comparisons (e-f). Given various signal to noise ratios, a higher revenue implies higher AARI values, that is better dendrograms.

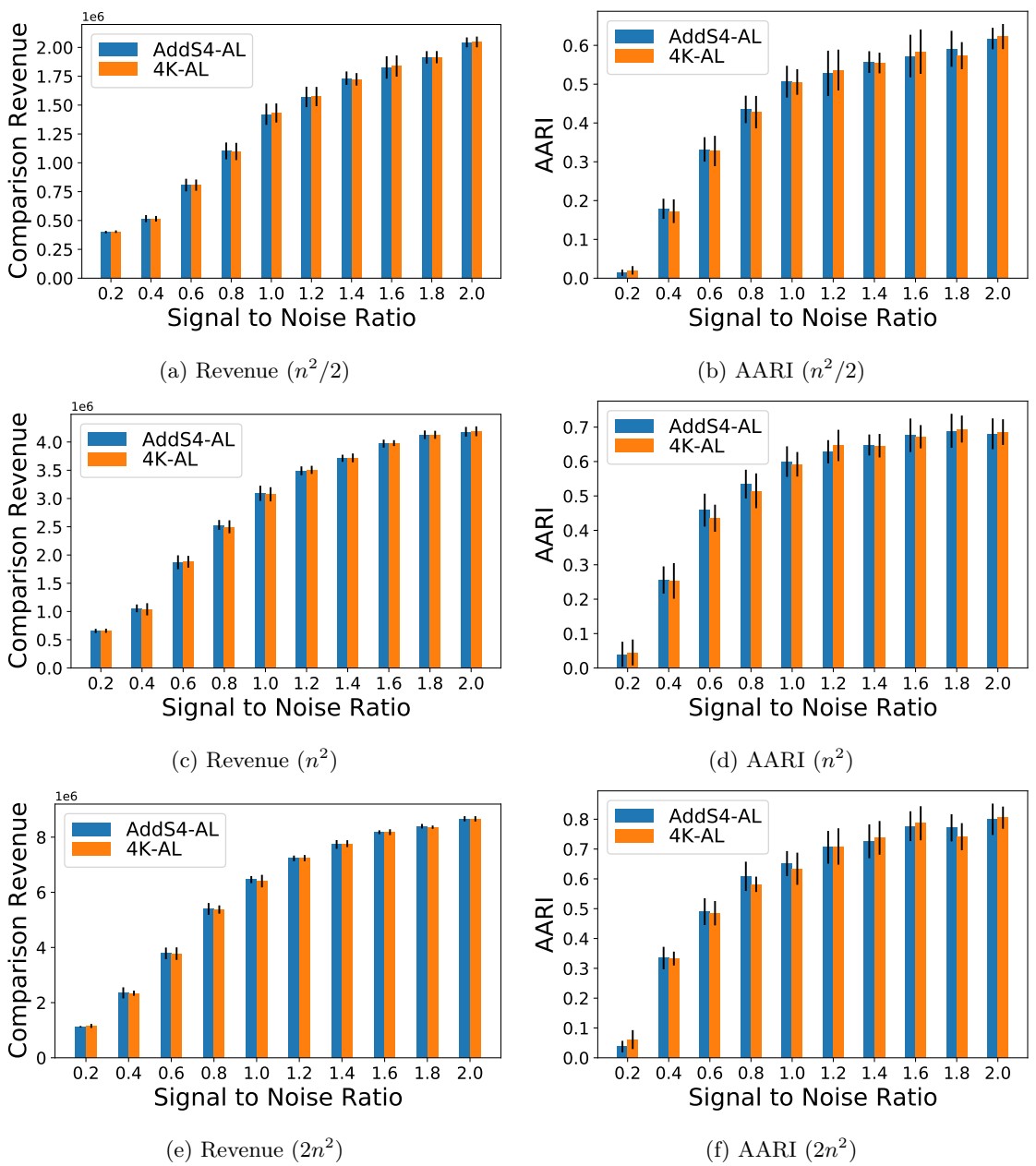

Figure 4: Revenue and AARI (higher is better) of several quadruplets-based methods using respectively $n^2/2$ comparisons (a-b), $n^2$ comparisons (c-d), and $2n^2$ comparisons (e-f). Given various signal to noise ratios, a higher revenue implies higher AARI values, that is better dendrograms.

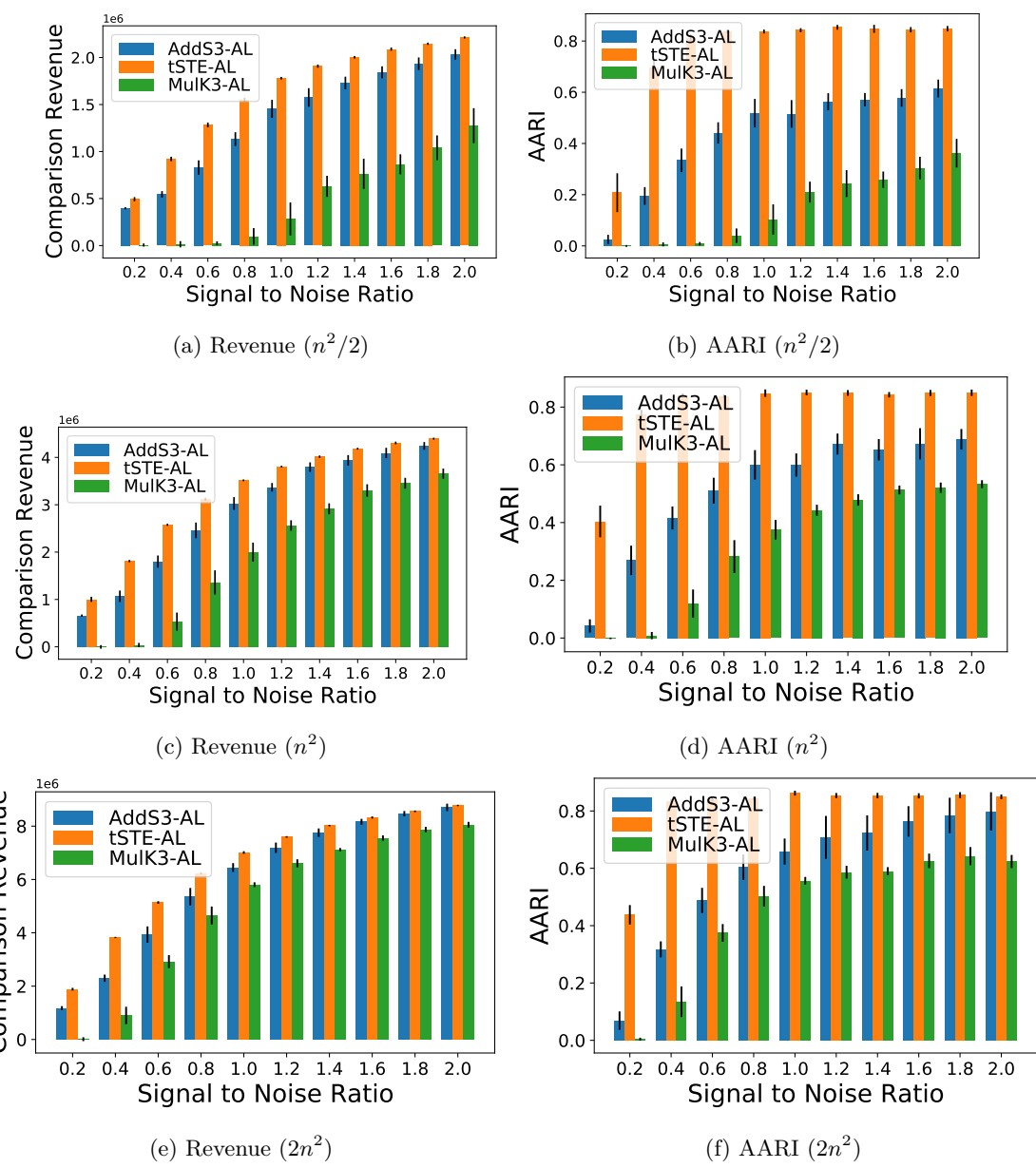

(a) Revenue $(n^2/2)$                 (b) AARI $(n^2/2)$

(c) Revenue $(n^2)$                  (d) AARI $(n^2)$

(e) Revenue $(2n^2)$                 (f) AARI $(2n^2)$

Figure 5: Revenue and AARI (higher is better) of several triplets-based methods using respectively $n^2/2$ comparisons (a-b), $n^2$ comparisons (c-d), and $2n^2$ comparisons (e-f) with 5% noise. Given various signal to noise ratios, a higher revenue implies higher AARI values, that is better dendrograms.

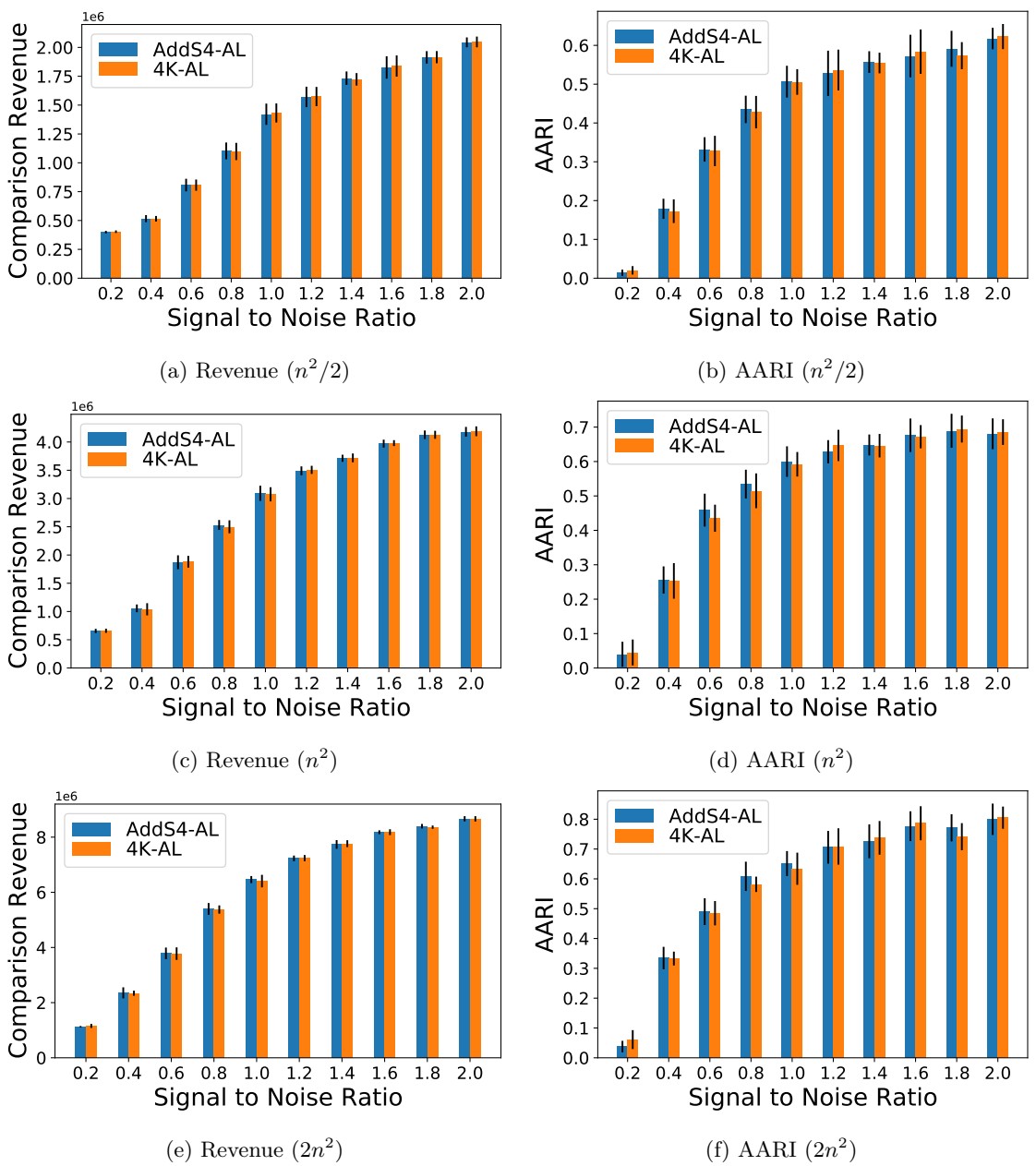

Figure 6: Revenue and AARI (higher is better) of several quadruplets-based methods using respectively $n^2/2$ comparisons (a-b), $n^2$ comparisons (c-d), and $2n^2$ comparisons (e-f) with 5% noise. Given various signal to noise ratios, a higher revenue implies higher AARI values, that is better dendrograms.

