# OpenReview forum: "A Revenue Function for Comparison-Based Hierarchical Clustering"
_TMLR — Accepted by TMLR_

### Review · Reviewer_etdf · 2023-02-17

**Summary Of Contributions:**

This work introduces a measure of quality for hierarchical clusterings on a set of objects. The novelty of this measure is that, instead of being based on pairwise similarities $s(A,B)$ between objects, it is based on comparisons of the kind "is $s(A,B)>s(A,C)$?". This is interesting since several clustering algorithms that are based on comparisons rather than similarities have been proposed; human themselves are often better at performing comparisons rather than providing similarities. After defining such a measure of quality, called revenue, the work relates it to other known cost functions and proves some interesting results. For instance, they prove that one can compute a hierarchy that has revenue within constant factors from the optimum if one samples roughly $O(n^2)$ of the $\Omega(n^3)$ possible triplets comparisons. The work is completed by some experiments that assess the effectiveness of the proposed measure in capturing the quality of a hierarchical clustering.



**Audience:**

Yes

**Broader Impact Concerns:**

I see no concern.

**Claims And Evidence:**

Yes

**Requested Changes:**

CRITICAL CHANGES

Page 5: "We propose to achieve this by finding a tree that maximises $Qrev(H, Q)$", how do you guarantee that in such a tree $|H(k \vee l)| > |H(i \vee j)|$? for all $(i,j,k,l) \in Q$? Moreover, the inequality could be loose (i.e., $|H(k \vee l)| = |H(i \vee j)|$).

Page 6: "one cannot exactly recover $H_0$ from any passively obtained $T \subseteq T_0$ if $|T| = o(n^3)$", this sentence is vague and unclear.

Page 6: "$i,j$ are merged before $k$", although I understand what is meant, there is no formal definition of what "merged before" means; and I do not think one can say "$i,j$ are merged before $k$" but rather "$i,j$ are merged before $i,k$ and $j,k$"

Page 6: "we obtain that $Trev(H_0, T_0) > Trev(H_0, T_1)$", couldn't this be an equality?

Page 7: "One can use standard concentration inequalities to show that, with high probability, $|T| = O(p_n n^3)$". This is true only if $p_n$ is large enough. In particular it is false if $p_n = n^{-3}$, since in that case $E[|T|]$ is too small to yield concentration.

Page 7, Theorem 3: "Given access to $T$", this has no meaning here.

Page 7, Theorem 3: "with probability at least $1-n^{-O(1)}$", probably you want to say $1-n^{-a}$ where $a>0$ can be made arbitrarily large by adjusting the constants. Otherwise that $O(1)$ could be $0$, in which case you get the trivial lower bound of $0$. In fact I would suggest defining formally what you mean by "high probability".

Page 7, Theorem 3: "Setting $p_n$ at its smallest allowable value, equation 5 holds for $|T| = O(n^2 \log n / \epsilon^2)$". Do you mean that, for that value of $p_n$, with high probability we have simultaneously $|T| = O(n^2 \log n / \epsilon^2)$ and equation (5)? I guess this is the intended meaning, but one could interpret that sentence differently -- for instance, that conditioned on the event $|T| = O(n^2 \log n / \epsilon^2)$ we have (5) with high probability. In any case, the sentence is missing some quantifiers and/or some probability.

Page 8: what do you mean by "ideal" matrix? I have not heard this before.

Page 8: so what objects are R,M precisely? Real symmetric matrices? This should be stated clearly and formally.

Page 8: "there are $2^L$ pure clusters", what is a cluster, and what is a pure cluster?

Page 9, Table 1: this experiment does not tell much. First, the range of values for the number of comparisons is rather small. I guess it would be better to test, say, from k=n/2^c to k=n in doubling steps. This would support the claim that your revenue is correlated with the AARI. Second, this test is basically anecdotal evidence. Better evidence would be given by computing some standard measure of correlation between revenue and AARI such as Pearson's correlation between the revenue and AARI sequences, or Spearman's rank correlation between the rankings of the hierarchies given by the revenue and the AARI for the various methods.

Page 10: the results are not easily repeatable since the authors used the current time to seed the random number generator (why not use just 0 as the seed?).

Page 10: the denominator of the cosine similarity should have the norms, and not the squared norms

Page 11: as before, these experiments are not repeatable since the authors used the current time to seed the random number generator.

Lemma 6: the statement is problematic as it first chooses $n, H_0, T_0$ and then makes a "universal" claim about all such choices. Consider rewriting it like this: "For every $\epsilon \in (0,1)$ there exists $n_0 > 0$ such that what follows holds for all $n \ge n_0$. If $H_0$ is a hierarchy on $[n]$ and $T_0$ is the set of triplets induced by $H_0$, then ...$"

Lemma 7: first, the $O(1)$ at the exponent of $n$ is not what you want, as I remarked above. Second, if you aim at $1-n^{-a}$ for arbitrarily large $a > 0$, then the hypothesis $p_n = \Omega(\log n / n)$ is not sufficient; you need the stronger hypothesis $p_n > c \log n / n$ for sufficiently large $c > 0$.




ORDINARY CHANGES

Page 3: "optimmisation" --> "optimisation"

Page 3: "an uniformly" --> "a uniformly",  "are" --> "is"

Page 3: "at least $\frac{1}{3}$-fraction of given triplets" --> "a fraction at least $\frac{1}{3}$ of the given triplets"

Page 3: "that satisfy at least $(1-\epsilon)$-fraction" --> "that satisfy at least a $(1-\epsilon)$-fraction"

Page 3: "given quartets" --> "the given quartets"

Page 4: "Since, we show that" --> "Since we show that"

Page 4: "we believe that ...": I do not understand this sentence; do you prove this is true, or what?

Page 4: "$C_1$ and $C_2$, denoting a split of $C$ such that ..." Why not just say that $\{C_1,C_2\}$ is a partition of $C$? Moreover, shouldn't the partition be nontrivial, i.e., $\min(|C_1|,|C_2|) > 0$ ?

Page 4: "..., or equivalently, the number of objects in the set $C$". I see no constraint in the definition that ensures that $|H(C)|=|C|$.

Page 4: "the number of objects in the set $C$", why not just $|C|$?

Page 4: "the smallest node in the tree and the smallest subtree containing both i and j" --> "the smallest node in the tree containing both i and j, and the smallest subtree containing both i and j"; otherwise it's ambiguous.

Page 4: "that allows ones" --> "that allows one"

Page 4: "o(n)" --> "o(\cdot)"

Page 4: "achieves $Drev(H, s) \ge \frac{1}{3}$-optimal revenue", what does this mean?

Page 5: "avoids counting the same comparison multiple times", what does this mean? I see no "counting" problem here.

Page 5: "in practice, the observed comparisons are fewer", in practice where? Moreover, this claim does not seem to be supported (there is no justification or citation).

Page 5: "We further assume that $T$, or $Q$, is passively collected -- the algorithm cannot decide which comparisons should be queried.", I am confused. What can the algorithm do? Query any triplet from $T$ where $T$ is an arbitrary subset of $T_{all}$?

Page 5: "which leads to $|H(i \vee k)| > |H(i \vee j)|$", you mean that you would like this to hold for every $(i,j,k) \in T$?

Page 5, Theorem 1: perhaps highlight in the notation that $s^{AddS3}$ is a function of the set of triplets $T$. Otherwise the right-hand side of the first equation looks independent of $T$, which is weird.

Page 5, Theorem 1: need brackets in the right-hand side of the equation of $s_{ij}^{AddS3}$

Page 6, Theorem 1: need brackets in the right-hand side of the equation of $s_{ij}^{AddS4}$

Page 6: "$Trev(H!,T_0)$"

Page 7: "that $\Omega(n^3)$ is necessary" --> "that $|T| \in \Omega(n^3)$ is necessary"

Page 8: "experimentsto"

Page 8: "triplets based" --> "triplets-based",  "quadruplets based" --> "quadruplets-based"

Page 8: "$M+R$ where," --> "$M+R$, where"

Page 8: "to be more difficult" --> "more difficult"

Page 8: "comparison based setting" --> "comparison-based setting"

Page 8: "well known measure" --> "well-known measure"

Page 8: "higher revenues imply higher AARI", you mean AARI against the optimal hierarchy?

Page 9: "several triplets (a-b) and quadruplets (c-d) based" --> "several triplet-based methods (a-b) and quadruplet-based methods (c-d)"

Page 10, Table 2: "triplets based" --> "triplet-based",  "quadruplets based" --> "quadruplet-based"

Page 10: "in this work to various" --> "in this work, to various"

Page 10: "quadruplets based" --> "quadruplet-based",  "triplets based" --> "triplet-based",  "similarity based" --> "similarity-based"

Page 10: "we can then use" --> "and then use"

Page 11: "comparison based" --> "comparison-based"

Page 11: "we randomly and uniformly flip 5% of the comparisons (Emamjomeh-Zadeh and Kempe, 2018), for example if one should observe the triple $(i, j, k)$ that is object $x_i$ is closer to $x_j$ than to $x_k$ then one observes $(i, k, j)$ instead", perhaps I would say "we randomly and uniformly flip 5% of the comparisons (Emamjomeh-Zadeh and Kempe, 2018), where by flipping $(i, j, k)$ we mean replacing it with $(i, k, j)$."

Page 11: "The number of objects, the kind of query" --> "The number of objects and the kind of query"

Page 11: it is not clear what kind of comparisons the real-data datasets contain, and how they are converted into triplets/quadruplets. For instance I wonder what the "most central triplet" is. I believe this should be explained in the main body of the manuscript.

Page 12: "two average linkage based algorithms" --> "two algorithms based on average linkage"
Proof of Theorem 1, Lemma 4, Lemma 5, Lemma 6, ... : the hierarchies $H$ are sometimes called "hierarchies", sometimes "binary trees", sometimes "trees".

Lemma 6: what is a "latent" hierarchy? It seems to me that it has no mathematical meaning.



**Strengths And Weaknesses:**

STRENGTHS

1) The problem is well-motivated. Several works exist that study comparison-based clustering, yet I am not aware of a measure of cluster quality that is defined in terms of comparisons.
2) The contributions are mostly clear and understandable.
3) The results and definitions are nicely connected to their related counterparts (e.g., Dasgupta's cost function for hierarchical clusterings).

WEAKNESSES

1) The work suffers from lack of formality and rigor in several points. This includes some sloppy or incomplete definitions, some slightly wrong formulation of hypotheses, and some slightly wrong formulation of bounds and results.
2) The sketches of the proofs are really too short, to the point that they do not manage to convey the idea, and therefore in the end they become quite useless.
3) The experimental part can be improved. On the one hand, there is no clear statement of what the whole experimental part is supposed to assess (the individual experiments are described, but here I'm talking about the general goal of the experiments). On the other hand, some experiments are not very solid and seem almost anecdotal (more about this below).

---

> ### Author Response · Authors · 2023-03-08
> **Response to comments of Reviewer etdf**
>
> 1. The work suffers from lack of formality and rigor in several points. This includes some sloppy or incomplete definitions, some slightly wrong formulation of hypotheses, and some slightly wrong formulation of bounds and results.
>
> Ans: We have taken into account reviewer's specific comments on this.
>
> 2. The sketches of the proofs are really too short, to the point that they do not manage to convey the idea, and therefore in the end they become quite useless.
>
> Ans: We provide elaborate proofs in our appendix. The goal of the proof sketches was to provide an high level idea of how we can arrive at the results (we see some value in this). If the reviewer still feels they are useless, we can remove them altogether.
>
> 3. The experimental part can be improved. On the one hand, there is no clear statement of what the whole experimental part is supposed to assess (the individual experiments are described, but here I'm talking about the general goal of the experiments). On the other hand, some experiments are not very solid and seem almost anecdotal (more about this below).
>
> Ans: Through the planted model experiment we tried to demonstrate the capability of AddS3-AL, tSTE-AL and MulK3-AL in finding the desired clusters. More generally, we also wanted to show the alignment between the AARI scores and Revenue values. This helps in validating the usefulness of the revenue values. In the second part of the experiments we show how revenue values can be utitlised in real world datasets. We have now made the objective clear at the beginning of the section.
>
> 4. Page 5: "We propose to achieve this by finding a tree that maximises QRev", how do you guarantee that in such a tree H(k ∨ l) > H(i ∨ j) for all (i,j,k,l) belongs to Q. Moreover, the inequality could be loose (i.e., H(k ∨ l) = H(i ∨ j)).
>
> Ans: We cannot provide any guarantee for this issue. More precisely, it may not be meaningful to compare $|H(i\vee j)|$ and $|H(k\vee l)|$ from two different parts of the tree, and hence, the worst case guarantee could be bad.
>
> 5. Page 6: "one cannot exactly recover H0 from any passively obtained T subset of T0 if |T|= o(n^3)", this sentence is vague and unclear.
>
> Ans: We could not clearly understand what is vague and unclear in this statement. We request the reviewer to be more precise on what aspect of the sentence is unclear.
>
> 6. Page 6: "i,j are merged before k", although I understand what is meant, there is no formal definition of what "merged before" means; and I do not think one can say "i,j are merged before k" but rather "i,j are merged before i,k and j,k"
>
> Ans: We have rewritten it more formally in the updated paper. In the updated version we mention $|H_0(I \vee j)| < \min(|H_0(I \vee k)|, |H_0(j \vee k)|)$ instead of "i,j are merged before k".
>
> 7. Page 7: "we obtain that $Trev(H_0,T_0)>Trev(H_0,T_1)$", couldn't this be an equality?
>
> Ans: No. We show in Lemma 5 that the optimal is unique.
>
> 8. Page 7: "One can use standard concentration inequalities to show that, with high probability, $|T| = O(p_n n^3)$". This is true only if $p_n$ is large enough. In particular it is false if $p_n = n^{-3}$ ,since in that case $E[|T|]$ is too small to yield concentration.
>
> Ans:  We state the condition on $p_n$ in the theorem statement.
>
> 9. Theorem 3: "Given access to T", this has no meaning here.
>
> Ans: We have removed this part in the updated version.
>
> 10. Theorem 3: "with probability at least $1-n^{-O(1)}$", probably you want to say $1-n^{-a }$ where $a>0$ can be made arbitrarily large by adjusting the constants. Otherwise that $O(1)$ could be 0, in which case you get the trivial lower bound of 0. In fact I would suggest defining formally what you mean by "high probability".
>
> Ans: We have made the theorem statement precise to include the probability.
>
> 11. Page 7, Theorem 3: "Setting $p_n$ at its smallest allowable value, equation 5 holds for $|T| = O(n^2log(n)/\epsilon^2)$". Do you mean that, for that value of $p_n$, with high probability we have simultaneously $|T| = O(n^2log(n)/\epsilon^2)$ and equation (5)? I guess this is the intended meaning, but one could interpret that sentence differently -- for instance, that conditioned on the event $|T| = O(n^2log(n)/\epsilon^2)$ we have (5) with high probability. In any case, the sentence is missing some quantifiers and/or some probability.
>
> Ans: The theorem is rephrased.
>
> 12. Page 8: what do you mean by "ideal" matrix? I have not heard this before.
>
> Ans: We have rephrased it in the updated version. We have replaced the term ideal symmetric matrix by real symmetric matrix.
>
> 13. Page 8: so what objects are R,M precisely? Real symmetric matrices? This should be stated clearly and formally.
>
> Ans: We have dropped R (to reduce notations) and stated M more formally and precisely in the updated version of our paper.

---

> > ### Comment · Reviewer_etdf · 2023-03-11
> > **Re:**
> >
> > *Ans: We provide elaborate proofs in our appendix. The goal of the proof sketches was to provide an high level idea of how we can arrive at the results (we see some value in this). If the reviewer still feels they are useless, we can remove them altogether.*
> >
> > It's clear what the goal of the sketches is. My point is precisely that the current sketches do not achieve fully that goal. They should convince me that the (full) proof works, without providing all details, but they don't. For instance, the proof sketch of Proposition 2 merely states two intermediate results that together yield the claim, but it does not give an intuition of why those results hold. The same holds for the proof of Theorem 3, which states three results but gives no intuition about them. So I find that those sketches actually bring more questions than answers.
> >
> >
> > *Ans: We could not clearly understand what is vague and unclear in this statement. We request the reviewer to be more precise on what aspect of the sentence is unclear.*
> >
> > First, "passive" is not defined. Second, the statement has several possible interpretations:
> >
> > &nbsp;&nbsp; - every passive algorithm that learns $o(n^3)$ triplets has a nonzero probability of failure
> >
> > &nbsp;&nbsp; - every passive algorithm that learns $o(n^3)$ triplets has a large probability of failure
> >
> > &nbsp;&nbsp; - every passive algorithm that learns $o(n^3)$ triplets always fails
> >
> > The current sentence suggests the last statement, but that statement is clearly false.
> >
> >
> > *Ans: We state the condition on  $p_n$ in the theorem statement.*
> >
> > That is clear. My point is that *the sentence before that statement is false* since it does not put any condition on $p_n$.
> >
> >
> > *Ans: We have made the theorem statement precise to include the probability.*
> >
> > The new statement is somewhat better, but it seems to me that it still uses wrong quantifiers. It says that for any fixed $\epsilon > 0$ there exist $c,c' > 0$ such that when $p_n > c f(n)$ then something happens with probability $1-n^{-c'}$. That gives you no control over the failure probability; as far as we know, it could be that $c'=(\frac{\epsilon}{2})^{1000}$. The correct statement is probably that for any fixed $\epsilon > 0$ and $a > 0$ there exist $c=c(\epsilon,a)$ such that when $p_n > c f(n)$ something happens with probability $1-n^{-a}$. This means that if I want failure probability $1-n^{-3}$ I can achieve it by taking more samples.
> >
> > More in general, I still think that it would be better to define formally what "with high probability" means --- usually, it means that you can always achieve $1-n^{-a}$ for any desired $a > 0$, at the price of an $f(a)$ factor in some other cost, and where typically $f$ is a linear function.
> >
> > *Ans: The theorem is rephrased.*
> >
> > The rephrasing is still wrong: the new sentence does not mention any probability of failure.
> >
> > *Ans: We have dropped R (to reduce notations) and stated M more formally and precisely in the updated version of our paper.*
> >
> > I see no matrix $M$ in the new version. There is, like before, a matrix $S$, but it is not stated that it is symmetric. I also note that it is not said whether the entries $s_{ij}=s_{ji}$ are independent from all other entries (I guess they are).
> >
> >
> > *Ans: We pick the time as seed to avoid picking good results. We also provide the code for reproducibility. We believe that results reproducibility is more essential than repeatability. In particular, Table 5 in appendix (with fixed seeds) corresponds Table 1 (or its extended version Table 4, with current time seed). The results shows minor fluctuations due to random seeds. If the reviewer thinks that a fixed seed is absolutely neccessary, we can rerun our experiments with a fixed seed.*
> >
> > Using time() gives no guarantees at all against cherrypicking. You could simply re-run the experiments a thousand times using time() as seed, stopping as soon as you like the results. Using 0 removes this issue, and is not less repeatable/reproducible than using time().
> >
> >
> > *Ans: As noted for Theorem 3, the results are now rephrased to address this issue.*
> >
> > The lemma still has the same issues that I raised above for Thm 3.

---

> ### Author Response · Authors · 2023-03-08
> **continued from previous response**
>
> 14. Page 8: "there are $2^L$ pure clusters", what is a cluster, and what is a pure cluster?
>
> Ans: By pure cluster we wanted to state that there are $2^L$ ground truth clusters. We have rephrased this part in the updated version and only use the term "$2^L$ clusters" instead of "$2^L$ pure clusters".
>
> 15. Page 9, Table 1: this experiment does not tell much. First, the range of values for the number of comparisons is rather small. I guess it would be better to test, say, from $k=n/2^c$ to $k=n$ in doubling steps. This would support the claim that your revenue is correlated with the AARI.
>
> Ans: We have updated Table 1 to show doubling steps
>
> 15. Second, this test is basically anecdotal evidence. Better evidence would be given by computing some standard measure of correlation between revenue and AARI such as Pearson's correlation between the revenue and AARI sequences, or Spearman's rank correlation between the rankings of the hierarchies given by the revenue and the AARI for the various methods.
>
> Ans: We have added Figure 2, and stated Kendall and Spearman rank correlation coefficients.
>
> 16. Page 10: the results are not easily repeatable since the authors used the current time to seed the random number generator (why not use just 0 as the seed?).
>
> Ans: We pick the time as seed to avoid picking good results. We also provide the code for reproducibility. We believe that results reproducibility is more essential than repeatability.  In particular, Table 5 in appendix (with fixed seeds) corresponds Table 1 (or its extended version Table 4, with current time seed). The results shows minor fluctuations due to random seeds. If the reviewer thinks that a fixed seed is absolutely neccessary, we can rerun our experiments with a fixed seed.
>
> 17. Page 10: the denominator of the cosine similarity should have the norms, and not the squared norms
>
> Ans: We thank the reviewer for pointing out this typo. We have corrected this in our updated version of the paper. This was (fortunately) a typo and the implementations are correct.
>
> 18. Page 11: as before, these experiments are not repeatable since the authors used the current time to seed the random number generator.
>
> Ans: see 16.
>
> 19. Lemma 6: the statement is problematic as it first chooses n,H0,T0 and then makes a "universal" claim about all such choices. Consider rewriting it like this: "For every epsilon belongs to (0,1) there exists $n_0>0$ such that what follows holds for all $n>=n_0$. If H0 is a hierarchy on [n] and T0 is the set of triplets induced by H0, then ..."
>
> Ans: Thank you. We have now rephrased the statement.
>
> 20. Lemma 7: first, the O(1) at the exponent of n is not what you want, as I remarked above. Second, if you aim at $1-n^{-a}$ for arbitrarily large $a>0$, then the hypothesis $p_n = Omega(log n/n)$ is not sufficient; you need the stronger hypothesis $p_n>c log n/n$ for sufficiently large $c>0$.
>
> Ans: As noted for Theorem 3, the results are now rephrased to address this issue.

---

### Review · Reviewer_EQsT · 2023-02-24

**Summary Of Contributions:**

This paper studies hierarchical clustering in the comparison-based setting. That is rather than have direct access to pairwise similarities, we only have triplet (i more similar to j than k) or quadruplet comparisons (i and j more similar to each other than to k and l). The paper introduces an objective for comparison-based hierarchical clustering that is closely related to Dasgupta's cost. Furthermore, it presents principled algorithms along with an empirical comparison of these methods.

**Audience:**

Yes

**Broader Impact Concerns:**

No ethical concerns.

**Claims And Evidence:**

Yes

**Requested Changes:**

I only have a few minor writing comments:

Minor:
* "In the past decade, there has been an exponential growth in the scope of data science and machine learning." -> Personally, I think this reads as too broad a statement. I think that stronger engagement from the reader would be to lead with the "crowdsourcing and psychometrics" use case and give both applications of these and impacts of those applications as you do in the following paragraph.
* "those focusing on ordinal data analysis from crowd-sourced data Kleindessner and von Luxburg (2017); Ghoshdastidar et al. (2019)." -> citation parens
* " we believe that our revenue maximisation formulation can recover the latent hierarchy using only O(n^2 log n) comparisons under their latent model." -> would be nice to say a bit more here, perhaps turning it into a proper result (or describing more for future work).

**Strengths And Weaknesses:**

**Strengths**

* This is a very clearly written paper that provides an accessible and seemingly comprehensive description of the landscape of related work and where the proposed objective and algorithms fit in.
* The algorithms, while simple, are clearly motivated by the setting and serve for an empirical comparison to validate the algorithms and proposed objective.
* The authors provide connections between related algorithms, objectives and settings that are nice and could help inspire future work connecting these related areas.

**Weaknesses**
* I think the paper could benefit from a bit more discussion of: how comparison-based data comes to be and further discussion of considerations for such data. What is the scale of such datasets? What is the purpose of clustering in the example datasets (e.g., data exploration, some end-task, etc)? What are the characteristics of the clusterings desired for such data? Speed, accuracy, guarantees about the solution? The paper does a nice job placing itself in the related literature, I think it would benefit from this further discussion of the data.
* Similarly, I think it would be good to discuss the use of agglomerative clustering further, especially in relation to highly scalable variants such as those that operate on sparse graphs, e.g., [1,2] among others.

[1] Sumengen, Baris, et al. "Scaling hierarchical agglomerative clustering to billion-sized datasets." arXiv preprint arXiv:2105.11653 (2021).

[2] Dhulipala, Laxman, et al. "Hierarchical Agglomerative Graph Clustering in Poly-Logarithmic Depth." arXiv preprint arXiv:2206.11654 (2022).

---

> ### Author Response · Authors · 2023-03-08
> **Response to comments of Reviewer EQsT**
>
> - (weakness) I think the paper could benefit from a bit more discussion of: how comparison-based data comes to be and further discussion of considerations for such data. What is the scale of such datasets? What is the purpose of clustering in the example datasets (e.g., data exploration, some end-task, etc)? What are the characteristics of the clusterings desired for such data? Speed, accuracy, guarantees about the solution? The paper does a nice job placing itself in the related literature, I think it would benefit from this further discussion of the data.
>
> Ans: We refer to this in the introduction. Also the scale of the benchmark datasets can be found in Table 3.
>
> - (weakness) Similarly, I think it would be good to discuss the use of agglomerative clustering further, especially in relation to highly scalable variants such as those that operate on sparse graphs, e.g., [1,2] among others.
>
> Ans: The direction of papers suggested by the reviewer focus on fast/scalable agglomerative clustering for large number of objects. Currently, comparison-based literature does not deal with too many objects, but the challenge is with the number of comparisons required and runtime with respect to number of comparisons. We raise the question of poly-time algorithms in the concluding remarks.
>
> - (requested change) "In the past decade, there has been an exponential growth in the scope of data science and machine learning." -> Personally, I think this reads as too broad a statement. I think that stronger engagement from the reader would be to lead with the "crowdsourcing and psychometrics" use case and give both applications of these and impacts of those applications as you do in the following paragraph.
>
> Ans: We have made the first sentences of the introduction more precise.
>
> - (requested change) " we believe that our revenue maximisation formulation can recover the latent hierarchy using only O(n^2 log n) comparisons under their latent model." -> would be nice to say a bit more here, perhaps turning it into a proper result (or describing more for future work).
>
> Ans: We have rephrased this text since the model in Perrot et al. (2020) assumes k planted clusters (not a hierarchy), and the theory would not be meaningful in their framework. We rather wanted to emphasize on a noise model in Perrot et al. (2020).
> We have now added Corollary 4, which shows that the tree recovery result in Theorem 3 still holds in the noisy setting.

---

> > ### Comment · Reviewer_EQsT · 2023-03-25
> > **Thank you for your response**
> >
> > Thanks to the authors for their response. Following resolution of the last comment by reviewer etdf, I would also lean towards acceptance.

---

### Review · Reviewer_Te1E · 2023-02-26

**Summary Of Contributions:**

This work proposes a new revenue function for comparison-based hierarchical clustering. There are many good theoretical contributions including the equivalence between the proposed function and Dasgupta's cost function, and provably approximate recovery of the hierarchy based on O(n^2 log n) sampled comparisons based on the proposed function. Evaluation on real datasets also reveals the effectiveness of the proposed function. Overall, I think this work makes valid contributions to the research on hierarchical clustering.

**Audience:**

Yes

**Broader Impact Concerns:**

I see no ethical concerns.

**Claims And Evidence:**

Yes

**Requested Changes:**

1. In sec 5, when T0 is defined, it is unclear "i,j are merged before k". "merged" is not defined. Suggest using a more formal way to define this.

2. Typo in the last line of the proof sketch for Prop.2: H! should be H1.

3. Real data experiments: To make the paper self-contained, I suggest adding descriptions of how the hierarchical structure is formulated rather than "use the same pre-processing...".

4. Please explain why the proposed algorithm is worse than tSTE-AL when the number of comparisons is less than 6n^2.

**Strengths And Weaknesses:**

Strengths:

1. The paper is generally well-written. The idea is very clear. The authors also do a good job to position this work in the context of related works, so the contributions are also presented clearly.
2. The notations and derivations are good and rigorous. I have roughly gone through the proof.
3. The experiments are extensive including synthetic datasets and real datasets, which demonstrates the effectiveness of the proposed revenue function.

Weaknesses:
1. The biggest weakness is the theoretical side of the algorithm for this revenue function. Although the sample complexity O(n^2 log n) has been proved to achieve good approximation, no poly-time approximation algorithm is provided for this revenue function.
2. There are some typos. Some parts of the paper are not self-contained. I will list some of them in "Requested Changes".

---

> ### Author Response · Authors · 2023-03-08
> **Response to comments of Reviewer Te1E**
>
> Weakness-1. The biggest weakness is the theoretical side of the algorithm for this revenue function. Although the sample complexity O(n^2 log n) has been proved to achieve good approximation, no poly-time approximation algorithm is provided for this revenue function.
>
> Ans: We mention this as an open problem in the conclusion section. We specifically ask whether a polynomial time (or even linear time) algorithm achieve a constant factor approximation of the triplet revenue.
>
>
> Weakness-2. There are some typos. Some parts of the paper are not self-contained. I will list some of them in "Requested Changes".
>
> Ans: We thank the reviewer for carefully going through the paper and pointing out some of the typos. They are corrected in the updated version of the paper.
>
> Requested change-1. In sec 5, when T0 is defined, it is unclear "i,j are merged before k". "merged" is not defined. Suggest using a more formal way to define this.
>
> Ans: We have rewritten it more formally in the updated paper. In the updated version we mention |H0(i ∨ j)| < min(|H0(i ∨ k)|, |H0(j ∨ k)|) instead of "i,j are merged before k".
>
> Requested change-3. Real data experiments: To make the paper self-contained, I suggest adding descriptions of how the hierarchical structure is formulated rather than "use the same pre-processing...".
>
> Ans: This has been added to Section 7.2
>
> Requested change-4. Please explain why the proposed algorithm is worse than tSTE-AL when the number of comparisons is less than 6n^2.
>
> Ans: We don't have a clear answer for this since the behaviour of tSTE (or broadly ordinal embedding) has not been studied yet under planted models. It is likely that the tSTE embedding identifies the ground clusters better even with less triplets. However, the observation is not consistent with real data, and hence, we do not comment on this in the paper.

---

> > ### Comment · Reviewer_Te1E · 2023-03-12
> > **No further comments**
> >
> > Thanks for the response from the authors. I am okay with the response and have no further comments.

---

### Decision · Action_Editors · 2023-03-27

**Recommendation:** Accept as is

**Comment:**

After some revisions during the review process, the paper is now clearly written and would be a valid contribution to TMLR.

**Audience:**

The paper proposes a new utility function for hierarchical clustering that relies only on outcomes of triplet comparisons, without the need for a similarity function. This contribution is both well motivated theoretically and useful for practical purposes. It is thus expected to be of interest to many in the TMLR audience.

**Claims And Evidence:**

The paper is well-written, and provides rigorous definitions and proofs. Extensive experiments are also included, which support the claims that the proposed revenue function is helpful in practice. The paper also provides good context for its relation to previous works.